# Weakly-Supervised 3D Referring Expression Segmentation

## Abstract

3D Referring Expression Segmentation (3D-RES) aims to generate precise segmentation masks for targets based on free-form text descriptions. Despite significant advancements, current methods still rely on costly point-level mask-description pair annotations. In this paper, we introduce the Multi-Expert Network (MEN), a novel weakly supervised framework that utilizes the multimodal alignment of vision-language models across various semantic cues to reveal the relationships between descriptions and 3D instances. The primary challenges lie in effectively extracting and matching visual and textual context, while eliminating potential distractions. To address this, we propose the Multi-Expert Mining (MEM) and Multi-Expert Aggregation (MEA) modules. The MEM module employs multiple experts to extract semantic cues from full-context, attribute, and category dimensions. The MEA module mathematically consolidates the outputs of these experts, automatically assigning greater weight to more accurate ones, thus improving target selection accuracy and robustness. Extensive experiments on the ScanRefer and Multi3DRefer benchmarks demonstrate the effectiveness of our method in addressing the challenges of weakly supervised 3D-RES.

## 1 Introduction

3D Referring Expression Segmentation (3D-RES) has gained significant attention in the multimodal field, aiming to segment 3D objects by aligning referring expressions with point clouds (Qian et al., 2024; He & Ding, 2024; He et al., 2024; Huang et al., 2021; Wu et al., 2024b;a). Unlike 3D referring expression comprehension (Luo et al., 2022; Wang et al., 2023a; Chen et al., 2022; He et al., 2021; Feng et al., 2021; Zhao et al., 2021; Yuan et al., 2021; Cai et al., 2022), which only requires locating the target and generating a simple bounding box, 3D-RES involves producing an accurate segmentation mask for the target based on free-form text descriptions. This advancement is crucial for applications in human-computer interaction and autonomous driving.

Despite notable progress (Huang et al., 2021; Wu et al., 2024b; Qian et al., 2024), current methods still rely on costly point-level mask annotations for each description. For instance, annotating a scene in the ScanNet-v2 (Dai et al., 2017) dataset takes an average of 22.3 minutes. The labor-intensive nature of these annotations hinders the collection of large-scale 3D datasets, limiting the task's scalability to data-scarce domains. To address this, we propose a weakly supervised 3D-RES setting, where only 3D point clouds and referring expressions are provided without mask annotations, as shown in Fig. 1. In this setting, both 3D scenes and referring expressions are easier to obtain, eliminating the need for laborious point-level mask annotations.

One straightforward approach to address this challenge is to first utilize a pre-trained 3D model to extract proposals and render them into 2D images, which are then matched directly with referring expressions using well-established 2D image-text matching techniques such as CLIP (Radford et al., 2021). However, this approach isolates instances from their 3D scene context, thus failing to leverage the crucial contextual cues provided in the text for accurate target localization. Another approach involves selecting images with the highest visibility of 3D proposals from the video used to construct the 3D point cloud (Boudjoghra et al., 2024), and then employing CLIP for matching. While this method more effectively utilizes the contextual information from the text, it introduces a significant challenge: if the image contains multiple proposals, it can lead to considerable matching ambiguity.

Figure 1: (a) Fully supervised 3D-RES frameworks utilize point-level mask annotations as supervision. (b) Weakly supervised 3D-RES frameworks require only easily accessible annotations for training, such as class labels.

Thus, the main challenge of this task lies in how to efficiently mine and match both visual and textual context while filtering out potential distractions.

To achieve this goal, we propose a Multi-Expert Network (MEN), a framework that leverages multiple cues to accomplish text-based referential reasoning. This framework breaks the problem into two sub-tasks: first, how to effectively mine and match visual and textual context; second, how to integrate the results of the first step to filter out distracting information while highlighting the target instance. For the first sub-task, we introduce a Multi-Expert Mining (MEM) module, which designs experts to extract information from multiple semantic cues, specifically focusing on full-context, attribute, and category dimensions to derive the target's distribution. The full-context expert processes the complete visual and textual context, assessing the match between proposals and the description from the perspective of spatial relationships and overall scene. The attribute expert focuses on fine-grained details such as shape, color, and texture of the instance, distinguishing proposals at the attribute level. The category expert, meanwhile, concentrates on the semantic category match between the proposal and the text description, ensuring robust matching unaffected by noise. For the second sub-task, directly integrating the probability distributions produced by the experts through summation or pooling may seem intuitive, but it fails to effectively suppress noise and emphasize the target. Therefore, we propose a Multi-Expert Aggregation (MEA) module that consolidates the match distributions generated by the three experts through the framework of distribution conflation. This module automatically assigns greater weight to input distributions derived from more accurate experts, specifically those exhibiting smaller standard deviations. In contrast to straightforward integration methods, the MEA module effectively filters out distractions and enhances the significance of the proposal corresponding to the target object.

To sum up, the contributions of this paper are three-fold:

- We propose a weakly supervised 3D-RES task, and introduce the Multi-Expert Network (MEN), a framework that leverages multiple cues for referring expression reasoning, significantly reducing the need for costly mask-description pair annotations.

- We design the Multi-Expert Mining (MEM) module, which effectively extracts and aligns visual and textual context from three complementary cues. Additionally, we introduce the Multi-Expert Aggregation (MEA) module, which integrates the knowledge from multiple experts, filtering out interference while emphasizing the target instance.

- Extensive experiments on ScanRefer and Multi3DRefer datasets demonstrate the effectiveness of our proposed method in addressing weakly-supervised 3D-RES.

## 2 RELATED WORK

### 2.1 3D REFERRING EXPRESSION COMPREHENSION AND SEGMENTATION

The 3D-REC aims to locate a target object in a 3D scene based on a referring language description and generate a bounding box (Luo et al., 2022; Wang et al., 2023a; Chen et al., 2022; He et al., 2021; Feng et al., 2021; Zhao et al., 2021; Yuan et al., 2021; Cai et al., 2022), while 3D-RES focuses on producing a corresponding segmentation mask (Qian et al., 2024; He & Ding, 2024; He et al., 2024; Huang et al., 2021; Wu et al., 2024b;a). The primary benchmark datasets for 3D-REC

and RES include ScanRefer (Chen et al., 2020) and ReferIt3D (Achlioptas et al., 2020). Recent approaches have started to explore weakly supervised settings for 3D-REC (Wang et al., 2023b; Xu et al., 2023). However, 3D-RES remains in its infancy, with current advancements primarily concentrated on fully supervised methods that necessitate point-level annotations. To reduce the reliance on costly annotations, we pioneer a weakly-supervised framework tailored for 3D-RES.

## 2.2 WEAKLY-SUPERVISED REFERRING IMAGE SEGMENTATION

Similar to 3D-RES, Referring Image Segmentation (RIS) aims to segment objects in images based on linguistic descriptions. Recent approaches (Hu et al., 2016; Yang et al., 2022; Ye et al., 2019; Li et al., 2018; Liu et al., 2023d; Yang et al., 2021; Liu et al., 2023b; Hu et al., 2023; Huang et al., 2020; Liu et al., 2023a; Xia et al., 2024; Liu et al., 2022) have achieved significant performance improvements in fully supervised settings. However, compared to fully supervised RIS, weakly supervised RIS (Lee et al., 2023; Dai & Yang, 2024; Kim et al., 2023; Liu et al., 2023c) is more challenging due to the lack of pixel-level annotations. TSEG (Strudel et al., 2022) introduces a weakly supervised setting to the RIS task for the first time. PPT (Dai & Yang, 2024) effectively leveraging SAM's (Kirillov et al., 2023) segmentation capabilities to obtain high-quality segmentation masks.

## 2.3 CONTRASTIVE LANGUAGE-IMAGE PRE-TRAINING

CLIP (Radford et al., 2021) is a multimodal model based on contrastive learning, trained on a large dataset of images and corresponding textual descriptions, which integrates visual and language modalities. It has been demonstrated to solve various tasks, such as open vocabulary segmentation (Takmaz et al., 2023; Yang et al., 2024; Nguyen et al., 2024). Although CLIP is widely used, its restricted input text length renders it inadequate for processing detailed descriptions. To address this issue, Zhang et al. (2024) proposed Long-CLIP as a plug-and-play alternative to CLIP, which supports longer text inputs while aligning with CLIP's latent space. In this paper, we introduce Long-CLIP processing of detailed descriptions to achieve a more comprehensive understanding.

## 2.4 OPEN-VOCABULARY 3D SEGMENTATION

Open-Vocabulary 3D Segmentation aims aims to extend the scope of 3D instance segmentation to any given short category name. OpenScene (Peng et al., 2023) leverages 2D open-vocabulary semantic segmentation models to map pixel-wise 2D CLIP features into 3D space. OpenMask3D (Takmaz et al., 2023) leverages SAM (Kirillov et al., 2023) and CLIP (Radford et al., 2021) to generate 3D clip features for each class-agnostic 3D proposal using RGB-D images of the 3D scene. Open3DIS (Nguyen et al., 2024) integrates 2D and 3D proposals using a novel fusion approach for 2D masks through hierarchical agglomerative clustering. Unlike Open-Vocabulary 3D Segmentation, which primarily emphasizes the extraction of visual features, 3D-RES often focuses on the alignment between complex textual descriptions and visual information.

# 3 METHOD

## 3.1 PROBLEM DEFINITION

Given a point cloud scene containing $N_p$ points, represented as $P \in \mathbb{R}^{N_p \times (3+F)}$, where each point has its 3D coordinates and $F$-dimensional auxiliary information, such as RGB. Correspondingly, given a text description containing $N_w$ words, represented as $\mathcal{T} \in \mathbb{R}^{N_w}$, which describes the object of interest. The goal of the 3D-RES task is to locate the target object based on the text and output the corresponding segmentation mask $M \in \{0, 1\}^{N_p}$. In the weakly-supervised setting, the training phase does not provide description-mask pairs for supervision, but only the class annotations $Y_{cls} \in \mathbb{R}^{N_{cls}}$ of the target objects, where $N_{cls}$ is the number of classes. Thus, we model the selection of the target object as solving for the discrete probability distribution $Q$ of the instance containing the target. By sampling from $Q$, we obtain the final predicted instance.

## 3.2 OVERVIEW

The framework of our proposed Multi-Expert Network (MEN) is depicted in Fig.2. Our model follows a segmentation-then-matching two-stage pipeline. In the first stage, we employ a pre-trained

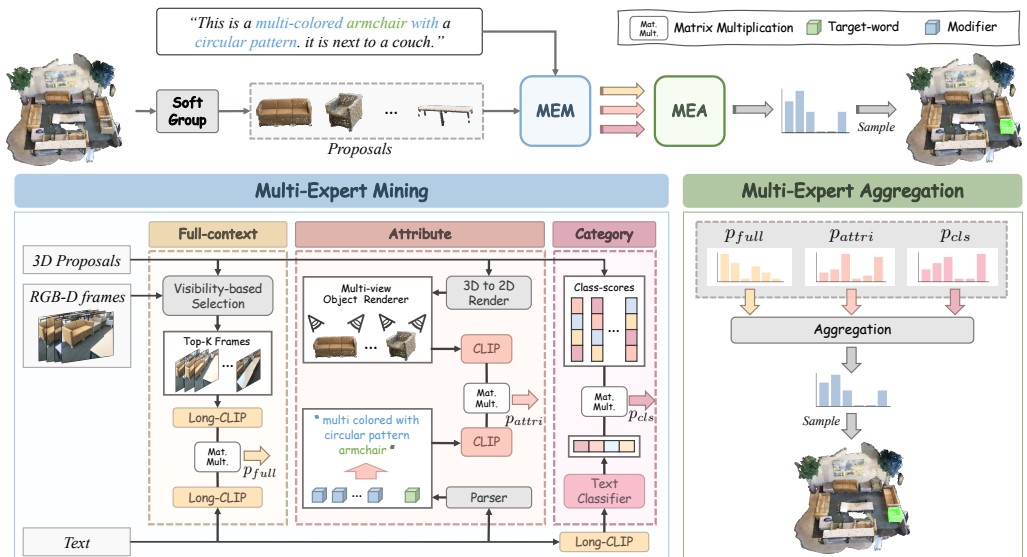

Figure 2: The overall framework of MEN, comprising MEM module and MEA strategy. After extracting mask proposals by a pre-trained instance segmentation model, the MEM module leverages multiple experts to extract the matching distributions of visual-language contexts. Finally, the MEA module efficiently consolidates these distributions, sampling to obtain the predicted target.

instance segmentation model to extract $N_o$ proposals from the given point cloud scene $P$, which can be represented as instance masks $\mathcal{M} \in \mathbb{R}^{N_o \times N_p}$ and class scores prediction $P_{cls} \in \mathbf{R}^{N_o \times N_{cls}}$. In the second stage, we introduce a Multi-Expert Mining (MEM) module that utilizes specialized experts to extract information from multiple semantic cues, focusing specifically on full-context, attribute, and category dimensions to derive the target's distribution, as detailed in Sec. 3.3. Finally, we consolidate the information from these experts using a Multi-Expert Aggregation (MEA) module, which efficiently filters out distractions while reinforcing the significance of the proposal corresponding to the target, as described in Sec. 3.4.

## 3.3 MULTI-EXPERT MINING

### 3.3.1 FULL-CONTEXT EXPERT

The full-context expert focuses on utilizing visual-language matching techniques to extract the complete visual-language context, understanding the spatial relationships and the surrounding environment of the target. However, text descriptions often include other objects in addition to the target and their spatial relationships with it, making the semantics complex and lengthy. As a result, CLIP struggles with such complex, lengthy texts (Radford et al., 2021).

Therefore, we utilize Long-CLIP (Zhang et al., 2024) on the textual side to comprehend the complete descriptive text. On the visual side, to capture comprehensive visual context, we select K frames from the video used to construct the 3D point cloud based on visibility.

Specifically, given the input description $\mathcal{T}$, we utilize the pre-trained Long-CLIP text encoder $\epsilon_{long}^{text}$ to extract textual full-context features $T_{full} \in \mathbb{R}^D$:

$$T_{full} = \epsilon_{long}^{text}(\mathcal{T}). \tag{3.1}$$

Then, given the $N_f$ RGB-D video frames $\mathcal{I} = \{\mathcal{I}_j \in \mathbb{R}^{4 \times W \times H} | \forall j \in (1, \ldots, N_f)\}$ associated with the 3D point cloud $P$, we follow Boudjoghra et al. (2024) to compute the visibility of 3D proposals across all frames in parallel. We select the Top-$K$ frames with the highest visibility for each 3D proposal as its visual representation, represented as $\mathcal{I}_{full} = \{\mathcal{I}_{full,i} \in \mathbb{R}^{K \times 3 \times W \times H} | \forall i \in (1, \ldots, N_o)\}$. These 2D frames will be encoded using Long-CLIP image encoder $\epsilon_{long}^{image}$, followed

by average pooling to obtain 2D full-context features $F_{full} \in \mathbb{R}^{N_o \times D}$:

$$F_{full} = \text{AvgPool}(\epsilon_{long}^{image}(\mathcal{I}_{full})), \tag{3.2}$$

where $\text{AvgPool}(\cdot)$ denotes the average pooling operation.

Finally, we compute the full-context level semantic similarity $\mathcal{S}_{full} \in \mathbb{R}^{N_o}$ and the normalized full-context semantic distribution $\mathcal{P}_{full}$ with probability mass function $p_{full} \in \mathbb{R}^{N_o}$:

$$\mathcal{S}_{full} = T_{full} \cdot F_{full}^T, \tag{3.3}$$

$$p_{full} = \text{Softmax}(\mathcal{S}_{full}), \tag{3.4}$$

where $T_{full} \in \mathbb{R}^D$ denotes textual full-context feature and $\text{Softmax}(\cdot)$ denote the softmax operation.

### 3.3.2 ATTRIBUTE EXPERT

Although the full-context expert effectively utilizes the visual-language context information, selected images often contain other proposals, leading to significant matching disturbances. In contrast, the attribute expert can thoroughly examine the instance's shape, color, texture, and other attributes, leveraging finer-grained cues for visual-language matching. To facilitate this, we specifically construct attribute text and attribute images as inputs for the attribute expert.

**Attribute Texts.** First, we extract key nouns and attribute terms through syntactic analysis. Following (Wu et al., 2023), we utilize the off-the-shelf natural language parser (Wu et al., 2019; Schuster et al., 2015) to syntactically parse the input description and decompose it into five distinct semantic components: `Main object` (the target object), `Auxiliary object` (which utilized to aid in identifying the main object), `Attributes` (objects' appearance, shape, etc.), `Pronoun` (refer to the primary object), `Relationship` (the spatial relation between main and auxiliary objects). We extract the main object and its corresponding attributes, represented as the target word $t_{main}$ and the modifiers $t_{mod}$, respectively. By concatenating these terms to create a short attribute phrase $\mathcal{T}_{short}$ that exclusively encompasses the name of the target object along with its modifiers, we subsequently encode this attribute phrase using a pre-trained CLIP text encoder $\epsilon^{text}$, thereby yielding the textual attribute features $T_{attri} \in \mathbb{R}^D$:

$$T_{attri} = \epsilon^{text}(\mathcal{T}_{short}). \tag{3.5}$$

**Attribute Images.** Meanwhile, we acquire the multi-view 2D renderings for each 3D proposal. Following Zhang et al. (2023), we send the proposals into an online object renderer, rendering multi-view 2D images for each 3D proposal, represented as $\mathcal{I}_{attri} = \{\mathcal{I}_{attri,i} \in \mathbb{R}^{K_v \times 3 \times W \times H} | \forall i \in (1, \ldots, N_o)\}$, where $K_v$ represents the number of views. Since the multi-view renderings for each 3D proposal contain only a single instance without any noisy background, they effectively highlight the attribute features of the instance. These images will be sent to the CLIP image encoder $\epsilon^{image}$ for feature extraction, followed by average pooling to obtain the 2D attribute features $F_{attri} \in \mathbb{R}^{N_o \times D}$:

$$F_{attri} = \text{AvgPool}(\epsilon^{image}(\mathcal{I}_{attri})), \tag{3.6}$$

where $\text{AvgPool}(\cdot)$ denotes the average pooling operation.

Finally, we compute the attribute level semantic similarity $\mathcal{S}_{attri} \in \mathbb{R}^{N_o}$ and the normalized attribute semantic distribution $\mathcal{P}_{attri}$ with probability mass function $p_{attri} \in \mathbb{R}^{N_o}$:

$$\mathcal{S}_{attri} = T_{attri} \cdot F_{attri}^T, \tag{3.7}$$

$$p_{attri} = \text{Softmax}(\mathcal{S}_{attri}), \tag{3.8}$$

where $T_{attri} \in \mathbb{R}^D$ denotes textual attribute feature and $\text{Softmax}(\cdot)$ denote the softmax operation.

### 3.3.3 CATEGORY EXPERT

To fully leverage the extracted 3D instance proposals and eliminate distractions, the category expert treats the predicted categories of proposals as visual priors while simultaneously extracting the category from the text for visual-language matching.

Specifically, we use an MLP as the textual classifier which receive the textual feature extracted from the textual backbone and output the textual class prediction scores $T_{cls} \in \mathbb{R}^{N_{cls}}$:

$$T_{cls} = \text{MLP}(T_{full}), \tag{3.9}$$

where the MLP($\cdot$) refers to text classifier, and $T_{full}$ denotes the feature of the textual description.

The text classifier is trained from scratch using a cross-entropy (CE) based classification loss $L_{cls}$:

$$L_{cls} = \text{CE}(T_{cls}, Y_{cls}), \tag{3.10}$$

where the $Y_{cls}$ is the category label of the target object. Then we compute the category level semantic similarity $\mathcal{S}_{cls} \in \mathbb{R}^{N_o}$ between the text and the proposals as follows:

$$\mathcal{S}_{cls} = T_{cls} \cdot P_{cls}^T, \tag{3.11}$$

where $P_{cls} \in \mathbf{R}^{N_o \times N_{cls}}$ class score predictions for 3D proposals generated by the instance segmentation model.

To more effectively model the semantic distribution $\mathcal{P}_{cls}$ with its probability mass function $p_{cls} \in \mathbb{R}^{N_o}$:

$$p_{cls} = \text{Softmax}(\mathcal{S}_{cls}), \tag{3.12}$$

where Softmax($\cdot$) denote the softmax operation.

### 3.4 MULTI-EXPERT AGGREGATION

The MEM module utilizes various experts to extract diverse and complementary visual-language cues, resulting in distinct discrete probability distributions: $P_{cls}$, $P_{attr}$, and $P_{full}$. However, integrating these distributions while eliminating distractions and emphasizing the target instance is a significant challenge. Thus, we first analyze the strengths and weaknesses of each expert.

**Strengths and Weaknesses of the Experts.** The full-context expert excels at capturing complete spatial context but can introduce distractions from complex 2D video frames, making it suitable for scenarios where spatial relationships are crucial for identifying targets. The attribute expert focuses on the visual attributes of the instance and the corresponding textual descriptions but may be influenced by rendering accuracy and parsing tools, making it ideal for cases with unique and distinct target attributes. The category expert leverages the classification capabilities of the visual backbone, demonstrating strong resistance to distractions by effectively eliminating irrelevant objects that do not match the target category, but it often lacks intra-class differentiation.

These experts exhibit different strengths, and their predicted distributions contain various distractions and conflicts, as illustrated in Fig. 3. Therefore, simply integrating these distributions through summation or pooling cannot effectively eliminate mutual interference, potentially undermining the enhancement of the final target distribution.

**Aggregation Approach.** To address this, we propose the Multi-Expert Aggregation (MEA) module, which mathematically assigns greater weight to the inputs from more accurate experts, specifically those with smaller standard deviations (Hill, 2011). The final distribution obtained through MEA is both the unique minimax likelihood ratio consolidation and the unique proportional likelihood ratio consolidation of the given input distributions (Hill, 2011), ensuring a robust integration of expert predictions.

Let the *conflation* of a finite number of probability distributions $\mathcal{P}_1, \ldots, \mathcal{P}_n$ be a consolidation of these distributions into a single probability distribution $Q = \&(\mathcal{P}_1, \ldots, \mathcal{P}_n)$, and designated with the symbol "&". According to Hill (2011), we can state

**Theorem 1.** *Let $\mathcal{P}_1, \ldots, \mathcal{P}_n$ be discrete with probability mass functions $p_1, \ldots, p_n$, respectively, and common atoms A, where $\emptyset \neq A \subset \mathbb{R}$. Then $\&(\mathcal{P}1, \ldots, \mathcal{P}n)$ exists, and the following are equivalent:*

(i) $Q = \&(\mathcal{P}1, \ldots, \mathcal{P}n)$

(ii) $Q = \frac{\sum_{x \in A} \delta_x \prod_{i=1}^n p_i(x)}{\sum_{y \in A} \prod_{i=1}^n p_i(y)}$

*where $\delta_x$ is the Dirac delta measure in the set of all real Borel probability measures at the point $x$ (i.e., $\delta_x(B) = 1$ if $x \in B$, and $= 0$ if $x \notin B$).*

Substituting $\mathcal{P}_{cls}$, $\mathcal{P}_{attr}$, and $\mathcal{P}_{full}$ yields:

$$Q = \frac{\sum_{x \in A} \delta_x p_{cls}(x) \cdot p_{attri}(x) \cdot p_{full}(x)}{\sum_{y \in A} p_{cls}(y) \cdot p_{attri}(y) \cdot p_{full}(y)}, \tag{3.13}$$

where common atoms $A$ here are the $N_o$ 3D proposals $\mathcal{M}$. Using the properties of Borel sets, here $\delta_x = 1$, thus this can be rewritten as a probability mass function $q$:

$$q = \frac{p_{cls} \cdot p_{attri} \cdot p_{full}}{\sum_{y \in \mathcal{M}} p_{cls}(y) \cdot p_{attri}(y) \cdot p_{full}(y)}, \quad (3.14)$$

where $\mathcal{M}$ denote the $N_o$ 3D proposals, $q \in \mathbb{R}^{N_o}$ denotes the probability mass function of the consolidation $Q$ of the multi-dimension semantic distributions. This aggregation method assigns greater weight to the inputs from more accurate experts (Hill, 2011), thereby adaptively eliminating distractions and reinforcing the significance of the target object.

Finally, we sample from $Q$ to predict the target instance $P_{res}$:

$$P_{res} = \text{Sample}(Q), \quad (3.15)$$

where $\text{Sample}(\cdot)$ denotes sample operation from a distribution.

## 4 EXPERIMENTS

### 4.1 DATASETS

**ScanRefer** (Chen et al., 2020) contains 51,583 natural language expressions of 11,046 objects from 800 ScanNet (Dai et al., 2017) scenes. It can be split into "Unique" and "Multiple", depending on whether there are multiple objects of the same category as the target in the scene.

### 4.2 EVALUATION METRIC

We adopted mIoU (mean Intersection over Union) and "R@$n$, IoU@$m$" as evaluation metrics, which represents the percentage of at least one of the top-$n$ predicted proposals with the highest probability having an IoU $> m$ compared to the ground truth mask, with $n \in \{1, 3\}$ and $m \in \{0.25, 0.5\}$. For the mIoU metric, we set $n = 1$.

### 4.3 EXPERIMENT SETTINGS

In our study, we utilize the pre-trained SoftGroup (Vu et al., 2022) as our 3D instance segmentation backbone. We freeze a pre-trained CLIP with ViT-B/32(Radford et al., 2021) and a pre-trained Long-CLIP with LongCLIP-L(Zhang et al., 2024). The text classifier is an MLP trained from scratch, starting with an initial learning rate of 0.001. To optimize this rate, we implement the Cosine Annealing strategy. It takes 20 epochs to train our model with a batch size of 32. The proposal number $N_o$, the number of rendering views $K_v$ are set to 256 and 6, respectively. For top-$K$ frames selection we use $K = 15$. To ensure reproducibility, our sampling procedure is configured for greedy sampling, wherein the target is selected from the top $n$ proposals with the highest probability mass, with $n \in \{1, 3\}$. All experiments are conducted on a single NVIDIA RTX 3090 GPU.

### 4.4 QUANTITATIVE COMPARISON

We present the results of our proposed MEN in comparison to existing 3D-RES models on ScanRefer, as shown in Tab. 1. For the sake of simplicity and efficiency, our weakly-supervised 3D-RES baseline is constructed based on the weakly-supervised 3D-REC model(Wang et al., 2023b), with modifications including changing the backbone to instance segmentation model SoftGroup (Vu et al., 2022) and excluding components related to word reconstruction and distillation. As shown in Tab. 1, the baseline achieved the expected performance, with a mIoU of 19.9%. We also randomly selected one target object from the same category based on the text class, denoted as "Baseline-random." Compared to the baseline, our model achieves improvements of 13.5% and 15.2% in the mIoU and "R@1, IoU@0.5", respectively, with a particularly notable increase of 13.9% on the more challenging "Multiple" subset. This demonstrates that our method not only excels at accurately identifying simple "Unique" targets but also effectively handles hard cases with multiple distracting objects similar to the target in the scene. This underscores how our MEM module thoroughly captures the complex semantics of both the scene and the descriptive text, while the MEA module efficiently consolidates these advantages, providing strong generalization and robust discriminative power. As a

Table 1: The 3D-RES results on ScanRefer. Fully-supervised models only have "R@1, IoU@$m$".

| Method | mIoU | R@1 | | | | | | R@3 | |
| | | Unique | | Multiple | | Overall | | Overall | |
| | | 0.25 | 0.5 | 0.25 | 0.5 | 0.25 | 0.5 | 0.25 | 0.5 |
|---|---|---|---|---|---|---|---|---|---|
| *Open-Vocabulary 3D segmentation Models* | | | | | | | | | |
| *OpenScene (Peng et al., 2023)* | *12.0* | *35.2* | *12.4* | *8.6* | *1.6* | *13.8* | *3.7* | *-* | *-* |
| *OpenMask3D (Takmaz et al., 2023)* | *12.8* | *20.8* | *19.8* | *13.9* | *12.4* | *15.2* | *13.8* | *-* | *-* |
| *Open3DIS (Nguyen et al., 2024)* | *12.3* | *22.3* | *11.9* | *13.8* | *5.9* | *15.4* | *7.1* | *-* | *-* |
| Fully-Supervised | | | | | | | | | |
| TGNN (Huang et al., 2021) | 27.8 | - | - | - | - | 37.5 | 31.4 | - | - |
| X-RefSeg3D (Qian et al., 2024) | 29.9 | - | - | - | - | 40.3 | 33.8 | - | - |
| 3D-STMN (Wu et al., 2024b) | 39.5 | 89.3 | 84.0 | 46.2 | 29.2 | 54.6 | 39.8 | - | - |
| RefMask3D (He & Ding, 2024) | 44.9 | 89.6 | 84.7 | 48.1 | 40.8 | 55.9 | 49.2 | - | - |
| MDIN (Wu et al., 2024a) | 48.3 | 91.0 | 87.2 | 50.1 | 44.9 | 58.0 | 53.1 | - | - |
| Weakly-Supervised | | | | | | | | | |
| Baseline-random | 17.7 | 59.5 | 54.7 | 14.0 | 11.6 | 22.8 | 19.9 | 36.2 | 31.8 |
| Baseline | 19.9 | 62.8 | 58.2 | 16.1 | 14.0 | 25.1 | 22.6 | 41.0 | 37.3 |
| Ours | 33.4 | 83.1 | 79.1 | **32.6** | **27.9** | **42.4** | **37.8** | **58.1** | **52.0** |
| Ours$_{STMN}$ | 29.6 | 84.7 | 79.1 | 30.2 | 18.8 | 40.7 | 30.5 | - | - |
| Ours$_{MDIN}$ | **34.5** | **87.3** | **82.6** | 31.4 | 26.9 | 42.3 | 37.7 | - | - |

result, our weakly supervised model significantly outperforms the fully supervised 3D-RES method TGNN (Huang et al., 2021) by 5.6 points on the mIoU metric.

We also applied open-vocabulary methods to the 3D-RES task. As shown in Tab. 1. These methods perform suboptimally, particularly in complex "multiple" setting, as they rely on single-prompt textual features from encoders such as CLIP, focusing on visual features while neglecting the complexities of hierarchical text descriptions and the fine-grained multimodal alignment required for 3D-RES. In contrast, our model addresses these limitations through the MEA module, achieving a 15.5% higher Acc@0.5IoU on "multiple" compared to OpenMask3D (Takmaz et al., 2023).

To directly extend weakly-supervised setting to fully supervised 3D-RES models, we also utilize MEN as a teacher model to generate pseudo-labels, *i.e.* target instance masks, to supervise the existing 3D-RES model. The results are denoted as Ours$_{ModelName}$. As demonstrated in Tab. 1, employing the pseudo-labels generated by MEN for supervision can assist common 3D-RES models in achieving competitive performance, even surpassing that of MEN.

## 4.5 ABLATION STUDY

**Ablation studies on MEM.** To investigate the effectiveness of MEM, we conducted an ablation study on the combinations of the three experts, as shown in Tab. 2. The full-context expert effectively mines spatial relationship information from the visual-linguistic context, enabling it to distinguish target objects from similar distractors. However, it faces challenges in the "Unique" category due to matching ambiguity when multiple instances are present within the selected 2D frames. In contrast, the category expert struggles to address "multiple"

Table 2: Ablation studies on MEM, where "full-cont." means full-context expert, "attri." means attribute expert, "cate." means category expert.

| | | full-cont. | attri. | cate. | mIoU | R@1, IoU@0.5 | | |
| | | | | | | Unique | Multiple | Overall |
|---|---|---|---|---|---|---|---|---|
| 1 | w/o MEM | ✓ | ✕ | ✕ | 23.4 | 51.2 | 19.8 | 25.9 |
| 2 | w/o MEM | ✕ | ✓ | ✕ | 10.8 | 23.2 | 9.5 | 12.2 |
| 3 | w/o MEM | ✕ | ✕ | ✓ | 25.6 | 78.8 | 17.4 | 29.3 |
| 4 | w/ MEM | ✓ | ✓ | ✕ | 26.2 | 58.6 | 22.4 | 29.4 |
| 5 | w/ MEM | ✓ | ✕ | ✓ | 32.9 | 78.3 | 27.1 | 37.0 |
| 6 | w/ MEM | ✕ | ✓ | ✓ | 26.4 | 78.0 | 18.8 | 30.3 |
| 7 | w/ MEM | ✓ | ✓ | ✓ | **33.4** | **79.1** | **27.9** | **37.8** |

situations. Even with the assistance of the attribute expert in identifying targets at a finer granularity, the attribute expert remains inadequate when faced with distractors that share identical attributes within the scene. The combination of the three can generate complementary advantages. On one hand, category and attribute expert aid the model in filtering out surrounding objects based on their category and attributes. On the other hand, full-context expert compensates for the deficiency of spatial relationship information in category and attribute expert, enabling the target object to stand out from distractors with the same category and attributes.

**Ablation studies on MEA of 3D-RES.** To analyze the effectiveness of the MEA, we conducted an ablation study on different consolidation strategies, as presented in Tab.3. We explored the compar-

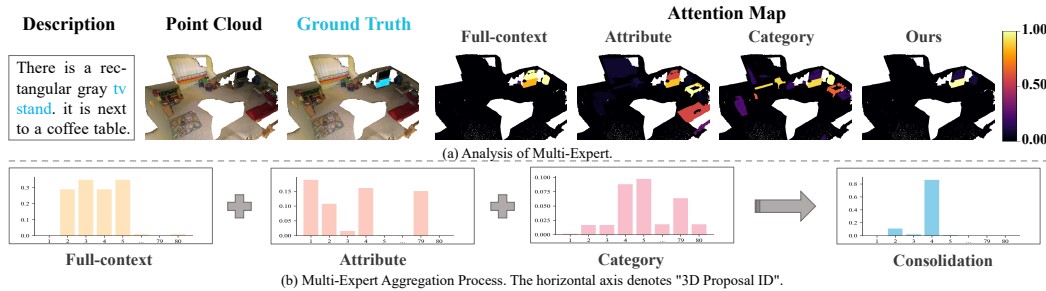

Figure 3: Visualization of **(a)** the attention maps and **(b)** the distributions outputted by experts.

ison between the "Max" and "Average" and our MEA. "Max" selects the highest probability from the three experts for each proposal as its corresponding probability, leading to a degradation of the model into a single expert mode and thereby suppressing the effect of multi-expert collaboration. "Average" calculates the average of the three probabilities and achieves good performance, but it fails to effectively leverage the strengths of the various experts and cannot filter out distractions. It can be demonstrated that the MEA effectively integrates the strengths of the three experts.

Table 3: Ablation studies on MEA.

| | Strategy | mIoU | Multiple | | Overall | |
|---|---|---|---|---|---|---|
| | | | 0.25 | 0.5 | 0.25 | 0.5 |
| 1 | Max | 26.6 | 22.6 | 19.6 | 33.2 | 30.1 |
| 2 | Average | 28.7 | 25.3 | 22.0 | 36.0 | 32.5 |
| 3 | MEA | **33.4** | **32.6** | **27.9** | **42.4** | **37.8** |

Table 5: Ablation studies on Long-CLIP in Full-context Expert.

| | Encoder | mIoU | Multiple | | Overall | |
|---|---|---|---|---|---|---|
| | | | 0.25 | 0.5 | 0.25 | 0.5 |
| 1 | SLIP (Mu et al., 2022) | 19.5 | 19.4 | 15.9 | 24.9 | 21.7 |
| 2 | CLIP (Radford et al., 2021) | 29.7 | 26.3 | 22.9 | 37.3 | 33.9 |
| 3 | Long-CLIP (Zhang et al., 2024) | **33.4** | **32.6** | **27.9** | **42.4** | **37.8** |

Table 4: Ablation studies on the image source.

| | Full-context | Attribute | mIoU | Multiple | | Overall | |
|---|---|---|---|---|---|---|---|
| | | | | 0.25 | 0.5 | 0.25 | 0.5 |
| 1 | Renderings | Renderings | 24.1 | 20.4 | 17.9 | 30.2 | 27.6 |
| 2 | RGB-D | RGB-D | 32.3 | 31.5 | 26.3 | 41.3 | 36.3 |
| 3 | Renderings | RGB-D | 26.1 | 22.6 | 19.5 | 32.9 | 29.7 |
| 4 | RGB-D | Renderings | **33.4** | **32.6** | **27.9** | **42.4** | **37.8** |

Table 6: Ablation studies on attribute phrase.

| | $\mathcal{T}_{short}$ | Prompt | mIoU | Multiple | | Overall | |
|---|---|---|---|---|---|---|---|
| | | | | 0.25 | 0.5 | 0.25 | 0.5 |
| 1 | Description | × | 33.0 | 32.4 | 27.6 | 42.2 | 37.4 |
| 2 | Target | ✓ | 32.9 | 31.9 | 27.3 | 41.8 | 37.3 |
| 3 | Target | × | 33.2 | 32.3 | 27.6 | 42.2 | 37.6 |
| 4 | Attribute | ✓ | 33.0 | 32.2 | 27.4 | 42.0 | 37.4 |
| 5 | Attribute | × | **33.4** | **32.6** | **27.9** | **42.4** | **37.8** |

**Image source in MEM.** We conducted an ablation study on the image source in MEM, as shown in Tab.4, to demonstrate the synergistic effect of multi-dimensional visual information. When both the full context and attribute expert are provided with renderings as input, the model lacks visual context information, leading to a significant reduction in performance. However, when both utilize RGB-D frames, the lack of individual object attribute analysis hinders the model's ability to focus effectively on a single object. And the results in the 3-rd row indicate that the mismatch between visual and linguistic information in experts can lead to a significant decline in performance.

**Long-CLIP in Full-context Expert.** We conducted an ablation study on the selection of Long-CLIP, as shown in Tab.5. Long-CLIP has been shown to possess a superior ability to comprehend long texts and capture image details, a conclusion that is also affirmed in this paper. Compared to directly utilizing CLIP, employing Long-CLIP results in a 3.7% improvement in mIoU.

**Text $\mathcal{T}_{short}$ in Attribute Expert.** We conducted an ablation study on the type of $\mathcal{T}_{short}$ to explore the role of attribute expert, As shown in Tab.6. Although the original description contains attributes of the target object, the properties of non-target objects may introduce interference when the description depends on other objects to aid in locating the target. Compared to using just the target word, incorporating attribute information results in more accurate localization. It is important to note that using the prompt templates provided by CLIP does not improve the model's performance.

## 4.6 PRINCIPLE ANALYSIS

In Fig.3, we illustrate the attention maps for Category, Attribute, and Full-context expert, along with the semantic distribution, to investigate their respective roles. As shown in **(a)**, these experts each serve a specific purpose, assisting the model in localizing targets at varying levels. The full-context

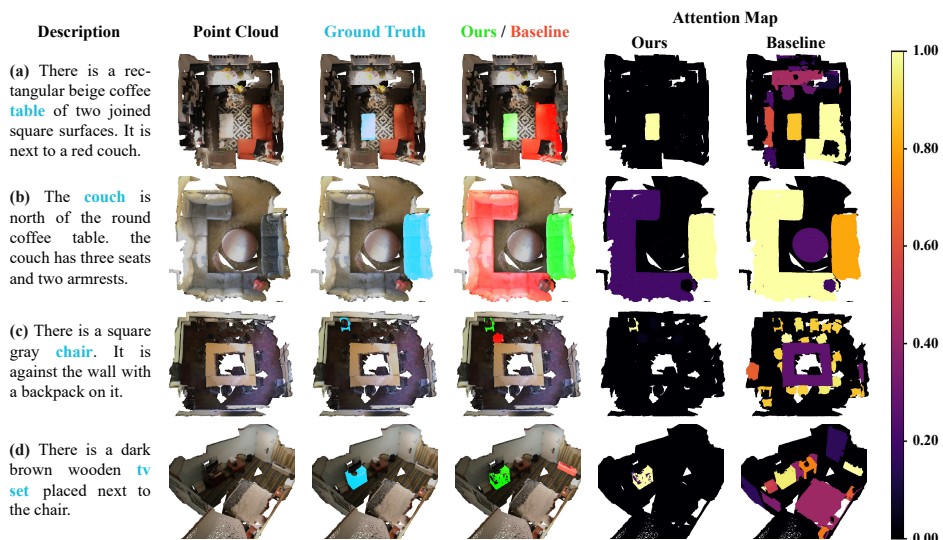

Figure 4: Visualization of the prediction results and attention maps of Ours and Baseline.

expert concentrate on scene corners relevant to the descriptionbut cannot distinguish between objects in those corners, such as the "TV," "TV stand," and the "blue object" shown in **(a)**. The attribute expert adhere to the indication of "rectangular gray TV stand," expressing interest in all objects that are "rectangular" "gray" or potentially classified as "TV stand." Similarly, the attention of the category expert is distributed across all objects in the scene that share a similar category to the target object. Therefore, while relying solely on the information of the target object is suitable for scenarios with unique categories and attributes, it proves ineffective in complex "multiple" scenarios. We can also observe that only objects exhibiting high similarity across all experts are likely to be predicted as targets. For example, the "blue object" near the target demonstrates high attention in both Category and Full-Context, but its low attention in Attribute diminishes its chances of being selected as the target. In summary, each expert possesses its own strengths and weaknesses. As shown in **(b)**, the MEA module leverages the complementary advantages of the three experts while filtering out distractions, thus enhancing the significance of the proposal for the target object.

## 5 QUALITATIVE COMPARISON

We present the visualization results of our MEN and the weakly-supervised baseline in Fig.4. Our model successfully identifies objects that match the target category from geometrically similar objects, as shown in **(a)**. Moreover, it can effectively utilize attribute information to distinguish between similar objects, such as the "three seats" depicted in **(b)**. Furthermore, it can infer the target object from multiple similar objects based on the full context, such as **(c)**. In contrast, the weakly-supervised baseline demonstrates an almost equal level of interest in all similar objects, which leads to inaccurate predictions. Impressively, as demonstrated in **(d)**, even when the target belongs to the "others" category, our model effectively eliminates distractions and achieves precise localization, aided by the experts and the efficient integration of the MEA module.

## 6 CONCLUSION

In this work, we introduce the Multi-Expert Network (MEN) for weakly supervised 3D Referring Expression Segmentation, eliminating the need for costly annotations. MEN consists of two modules: MEM and MEA. The MEM module employs multiple experts to extract semantic cues from various dimensions, while MEA robustly consolidates their outputs. Experiments demonstrate the effectiveness of our method in addressing the challenges of weakly supervised 3D-RES.

## ETHICS STATEMENT

We follow the ICLR Code of Ethics. In our work, there are no human subjects and informed consent is not applicable. For point cloud data, we used the publicly available ScanNet Dataset (`https://github.com/ScanNet/ScanNet`), which is licensed under the ScanNet Terms of Use and the code is released under the MIT license. Both licenses permit the use of the dataset and code strictly for non-commercial research purposes. Furthermore, we utilize the publicly accessible text data from the ScanRefer Dataset (`https://daveredrum.github.io/ScanRefer`), which is licensed under a *Creative Commons Attribution-NonCommercial-ShareAlike 3.0 Unported License* which allows us to use the dataset for non-commercial purposes. For 3D-GRES, We use the publicly accessible text data from the Multi3dRefer Dataset (`https://3dlg-hcvc.github.io/multi3drefer/#/`) which is licensed under the MIT license, permitting us to use the dataset for non-commercial purposes. In the appendix, we use the ReferIt3D Dataset (`https://github.com/referit3d/referit3d`) for extra experiments, which is licensed under the MIT license which allows us to use the dataset for non-commercial purposes.

## REPRODUCIBILITY STATEMENT

In accordance with the principles of reproducibility, we provide a comprehensive overview of the methodologies, experimental setup, code, and so on.

**Environment.** The experiments were conducted in a environment characterized by the following specifications:

- Operating System: Ubuntu 22.04.3 LTS
- Python Version: Python 3.8.13
- Framework: PyTorch 1.12.1

**Source Code Availability.** The complete code of our model is provided in the supplementary material and will be open-sourced. Additionally, we employs several pre-trained models, including SoftGroup, CLIP, and Long-CLIP:

- SoftGroup (`https://github.com/thangvubk/SoftGroup`).
- CLIP (ViT-B/32) (`https://github.com/openai/CLIP`).
- Long-CLIP (LongCLIP-L) (`https://github.com/beichenzbc/Long-CLIP`).

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

APPENDIX

## A  MORE QUANTITATIVE RESULTS

### A.1  DATASET

The ReferIt3D benchmark (Achlioptas et al., 2020) is constructed upon the ScanNet dataset (Dai et al., 2017) and comprises two primary datasets: Nr3D and Sr3D. Nr3D/Sr3D can be categorized into "easy" and "hard" subsets depending on the presence of multiple objects of the same target class within the scene. Based on whether the description depends on the observer's view, it can be further divided into "view-dep" and "view-indep" subsets.

**Nr3D** (Natural Reference in 3D) consists of 41.5K human descriptions collected using a referring game (Kazemzadeh et al., 2014). It describes objects in 707 ScanNet scenes.

**Sr3D** (Spatial Reference in 3D) contains 83.5K synthetic descriptions. It categorizes spatial relations into 5 types: horizontal proximity, vertical proximity, between, allocentric and support, and then generates descriptions using language templates.

### A.2  RESULTS

We compared the performance of our MEN with the existing 3D-RES models on the ReferIt3D benchmark in Tab. 8. Unlike the initial settings of ReferIt3D, we do not use the ground truth mask as input for proposals. Therefore, the overall performance of our model is constrained by the performance of the instance segmentation model. Nevertheless, our model still achieves considerable performance under weak supervision settings. Compared to the baseline, XX achieved improvements of 7.7% and 3.2% in mIoU on Nr3D and Sr3D, respectively. Notably, MEN surpassed the fully supervised TGNN by 2.7% in "R@1,IoU@0.5" on Nr3D.

### A.3  GENERALIZATION AND BACKBONE ANALYSIS

One concern raised was the potential data leakage due to the use of the SoftGroup instance segmentation backbone, which was trained on ScanNet, the source dataset for ScanRefer and Multi3DRefer. To further address this concern, we conducted additional experiments replacing the SoftGroup backbone with a model trained on the S3DIS dataset (Armeni et al., 2016). As shown in Table 7, our method consistently improved mIoU by 13.9% over the baseline, even with this alternative backbone.

Table 7: 3D-RES results on ScanRefer, where the instance segmentation models for both the baseline and our MEN are pre-trained on the S3DIS dataset, but evaluated on the ScanNet scenes.

| Method | mIoU | R@1 | | | | | |
| | | Unique | | Multiple | | Overall | |
| | | 0.25 | 0.5 | 0.25 | 0.5 | 0.25 | 0.5 |
|--------|------|------|-----|------|-----|------|-----|
| Baseline | 10.6 | 33.1 | 28.6 | 9.8 | 6.9 | 14.3 | 11.1 |
| Ours | **24.5** | **59.7** | **52.1** | **27.4** | **19.9** | **33.7** | **26.2** |

with this alternative backbone. This demonstrates the robustness of our approach and its independence from the specific training data of the original backbone. These results affirm the generalizability and flexibility of our method, supporting its applicability in unseen environments.

## B  MORE ABLATION STUDIES

**Top-$K$ Frames Selection in Full-context Expert.** In the full-context expert, the RGB-D video frames provide visual context. We conducted an ablation study on the choice of $K$ in the selection of Top-$K$ frames, as shown in Tab.9. As the value of $K$ increases, the model's performance exhibits a trend of initially rising and then declining, peaking at $K = 15$. We suspect that when too few frames are selected, the visual context contained is relatively limited. Instead, when too many frames are selected, objects that are physically close may correspond to multiple identical frames, leading to similar visual features and making them hard to distinguish.

**Rendering views $K_v$ in Attribute Expert.** We conducted an ablation study on the number of rendering views $K_v$, as presented in Tab.10. The number of rendering views is correlated with the quality of the 2D attribute features, which subsequently impacts semantic distribution from attribute

Table 8: The 3D-RES results on Referit3D benchmark, including mIoU and "R@1, IoU@$m$" with $m \in \{0.25, 0.5\}$.

| Method | mIoU | Easy 0.25 | 0.5 | Hard 0.25 | 0.5 | View-Dep 0.25 | 0.5 | View-Indep 0.25 | 0.5 | Overall 0.25 | 0.5 |
|---|---|---|---|---|---|---|---|---|---|---|---|
| Nr3D | | | | | | | | | | | |
| Fully-Supervised | | | | | | | | | | | |
| TGNN (Huang et al., 2021) | 19.1 | 29.2 | 22.3 | 22.5 | 19.6 | 22.2 | 18.1 | 27.6 | 22.4 | 25.7 | 20.9 |
| 3D-STMN(Wu et al., 2024b) | 27.6 | 47.9 | 31.9 | 35.4 | 20.0 | 37.7 | 21.1 | 43.5 | 28.4 | 41.5 | 25.8 |
| MDIN(Wu et al., 2024a) | 38.6 | 55.0 | 48.4 | 42.2 | 36.3 | 40.8 | 34.6 | 52.5 | 46.3 | 48.4 | 42.2 |
| Weakly-Supervised | | | | | | | | | | | |
| Basline | 13.5 | 21.7 | 19.0 | 12.4 | 11.2 | 16.9 | 14.6 | 16.9 | 15.2 | 16.9 | 15.0 |
| Ours | **21.2** | **31.5** | **27.9** | **23.4** | **19.6** | **24.7** | **20.2** | **28.8** | **25.5** | **27.3** | **23.6** |
| Sr3D | | | | | | | | | | | |
| Fully-Supervised | | | | | | | | | | | |
| TGNN (Huang et al., 2021) | 20.2 | 28.2 | 23.0 | 29.1 | 25.8 | 23.8 | 21.3 | 28.6 | 23.9 | 27.5 | 22.9 |
| 3D-STMN(Wu et al., 2024b) | 34.4 | 49.4 | 38.2 | 41.9 | 31.0 | 45.5 | 33.5 | 47.2 | 36.2 | 47.2 | 36.1 |
| MDIN(Wu et al., 2024a) | 46.4 | 58.9 | 53.2 | 51.1 | 46.8 | 53.6 | 48.7 | 56.7 | 51.4 | 56.6 | 51.3 |
| Weakly-Supervised | | | | | | | | | | | |
| Basline | 17.1 | 24.3 | 20.9 | 16.4 | 14.9 | 17.8 | 16.2 | 22.1 | 19.2 | 21.9 | 19.1 |
| Ours | **20.3** | **29.2** | **25.1** | **19.5** | **18.2** | **23.8** | **20.3** | **26.4** | **23.1** | **26.3** | **23.0** |

expert. As illustrated in Tab.10, the model's performance initially increases with the growth of $K_v$, reaching a peak at $K_v = 6$ before subsequently declining. Due to potential inaccuracies in instance segmentation results, having too few rendering views may lead to unrepresentative renderings. Conversely, an excessive number of rendering views can be redundant and may be adversely affected by by certain perspectives with poor visibility.

Table 9: Ablation studies on Top-$K$ Frames Selection in Full-context Expert.

| | Top-$K$ Frames | mIoU | Multiple 0.25 | 0.5 | Overall 0.25 | 0.5 |
|---|---|---|---|---|---|---|
| 1 | $K = 1$ | 30.7 | 29.3 | 25.1 | 38.9 | 34.9 |
| 2 | $K = 5$ | 32.5 | 31.4 | 26.9 | 41.2 | 36.8 |
| 3 | $K = 10$ | 33.1 | 32.4 | 27.6 | 42.2 | 37.6 |
| 4 | $K = 15$ | **33.4** | **32.6** | **27.9** | **42.4** | **37.8** |
| 4 | $K = 20$ | 33.0 | 32.2 | 27.3 | 42.1 | 37.4 |

Table 10: Ablation studies on rendering views in Attribute Expert.

| | Number of views | mIoU | Multiple 0.25 | 0.5 | Overall 0.25 | 0.5 |
|---|---|---|---|---|---|---|
| 1 | $K_v = 1$ | 32.4 | 31.9 | 27.1 | 41.5 | 36.8 |
| 2 | $K_v = 3$ | 33.0 | 32.2 | 27.4 | 42.0 | 37.4 |
| 3 | $K_v = 6$ | **33.4** | **32.6** | **27.9** | **42.4** | **37.8** |
| 4 | $K_v = 12$ | 32.2 | 32.4 | 27.6 | 42.1 | 37.6 |

Table 11: Ablation studies on 3D instance segmentation model.

| | Method | mIoU | Multiple 0.25 | 0.5 | ]Overall 0.25 | 0.5 |
|---|---|---|---|---|---|---|
| 1 | SoftGroup | 33.4 | 32.6 | 27.9 | 42.4 | 37.8 |
| 2 | SPFormer | 34.2 | 31.2 | 26.8 | 41.0 | 36.7 |
| 3 | GT Ins. | 48.1 | 38.1 | 38.1 | 48.1 | 48.1 |

Table 12: Ablation studies on the method of extracting $\mathcal{T}_{short}$.

| | Extracting Method | mIoU | Multiple 0.25 | 0.5 | Overall 0.25 | 0.5 |
|---|---|---|---|---|---|---|
| 1 | Parser | 33.4 | 32.6 | 27.9 | 42.4 | 37.8 |
| 2 | LLM | 33.5 | 32.7 | 27.9 | 42.6 | 37.9 |

**Ablation of 3D Instance Segmentation Models** We conducted ablation studies to evaluate the impact of different 3D instance segmentation models on the overall performance of our framework and explored the performance upper bound by eliminating errors in the one-stage instance proposal extraction, as shown in Table 11. The results indicate that the substitution of different instance segmentation models has little effect on the performance of our model, which further underscores its robustness.

**Ablation of the method of extracting attribute-level text** $\mathcal{T}_{short}$ In addition to using a traditional off-the-shelf natural language parser to extract attribute-level text, we also employed a large language model (LLM) for the same task. Specifically, we utilized GPT-3.5-turbo to generate attribute phrases, with the input prompts shown in Lst. 1. The results shown in Table 12 indicate that while LLMs are capable of generating more accurate attribute phrases, their impact on the overall model performance remains minimal.

Listing 1: Prompt used for extracting attribute-level phrases using GPT-3.5-turbo.

```
{
    "role": "system",
    "content":
        "You are an intelligent chatbot designed to extract attribute
            information of a target object from a given sentence."
        "Your task is to analyze the sentence I provide, which describes
            a target object, and distill it into a concise descriptive
            phrase containing the target object's name and its attributes
            ."
        "Note that all attribute words must be directly taken from the
            input description without alteration, omission, or addition."
        "Furthermore, you must not use attribute words unrelated to the
            target object in the description."
},
{
    "role": "user",
    "content":
        "Please extract the target and its attributes from the following
            input text and output a concise phrase."
        "Input sentence: {text}\n"
        "DO NOT PROVIDE ANY OTHER OUTPUT TEXT OR EXPLANATION. Only
            provide the target attribute phrase."
        "------"
        "Here are some examples:"
        "Input sentence: There is a brown wooden chair. Placed beside
            other chairs in the middle of the kitchen."
        "Your response should look like this: A brown wooden chair."
        "Input sentence: The couch is north of the round coffee table.
            the couch has three seats and two armrests."
        "Your response should look like this: A couch with three seats
            and two armrests."
        "Input sentence: There is a dark brown chair. brown leather and
            placed in the kitchen table."
        "Your response should look like this: A dark brown wooden and
            leather chair."
}
```

## C    DISCUSSION ON FRAME COVERAGE AND FULL-CONTEXT EXPERT

One concern is that the limited number of video frames selected by the full-context expert may result in cases where target objects and anchors are not present within the same frame due to the restricted field of view. To address this concern, we performed an analysis to assess the frequency and impact of such scenarios in the ScanRefer dataset.

Since the ScanRefer dataset lacks explicit annotations for all anchors and their presence in specific frames, we randomly sampled 100 examples from the validation set and conducted manual inspection. For each sampled example, we checked whether the 2D frames selected by our model contained all the target objects and anchors referenced in the text descriptions. The results showed that 95% of the sampled frames included all relevant objects, suggesting that the selected frames generally encompass the full visual context described in the text.

These findings align with the experimental results presented in Table 2, particularly in rows 2 and 4, where the inclusion of the *full-context expert* significantly improves performance. Specifically, adding the *full-context expert*, which captures spatial relationships and contextual information, results in a 15.4-point increase in mIoU compared to using only the *attribute expert*, which focuses solely on target-specific features.

While the aforementioned analysis indicates that this issue is relatively rare in the ScanRefer dataset, we recognize its importance and potential implications. Scenarios where some anchors are missing

Table 13: Results of 3D-GRES, where "zt w/ dis" means zero target with distractor, "zt w/o dis" means without distractor, "st w/ dis" means single target with distractor, "st w/o dis" means without distractor, "mt" means multiple target, "w/o zt_head" means that the zt_head was not used, and "w/ zt_head" means that the zt_head was used to predict whether it is a zero target.

| Method | mIoU | Acc@0.25 | | | | | | Acc@0.5 | | | | | |
|---|---|---|---|---|---|---|---|---|---|---|---|---|---|
| | | zt w/ dis | zt w/o dis | st w/ dis | st w/o dis | mt | Overall | zt w/ dis | zt w/o dis | st w/ dis | st w/o dis | mt | Overall |
| Fully-Supervised | | | | | | | | | | | | | |
| M3DRef-CLIP | 37.4 | 39.2 | 81.6 | 50.8 | 77.5 | 66.8 | 55.7 | 39.2 | 81.6 | 29.4 | 67.4 | 41.0 | 37.5 |
| 3D-STMN | 43.0 | 42.6 | 76.2 | 49.0 | 77.8 | 68.8 | 60.4 | 42.6 | 76.2 | 24.6 | 69.2 | 43.9 | 40.9 |
| MDIN | 47.5 | 47.9 | 78.8 | 55.5 | 84.4 | 76.3 | 67.0 | 47.9 | 78.8 | 29.5 | 71.7 | 46.8 | 44.7 |
| Weakly-Supervised | | | | | | | | | | | | | |
| Ours(w/o zt_head) | 30.6 | 1.3 | 0.4 | 41.8 | 67.7 | 47.7 | 44.8 | 1.3 | 0.4 | 29.8 | 59.5 | 19.8 | 30.1 |
| Ours(w/ zt_head) | 33.4 | 22.2 | 55.9 | 41.6 | 67.4 | 47.7 | 48.0 | 22.2 | 55.9 | 28.7 | 59.4 | 19.7 | 33.4 |

Table 14: Ablation results on thresholds and zero-target detection head, where "w/o zt_head" means that the zt_head was not used, and "w/ zt_head" means that the zt_head was used to predict whether it is a zero target.

| Setting | Threshold | mIoU | Acc@0.25 | | | | | | Acc@0.5 | | | | | |
|---|---|---|---|---|---|---|---|---|---|---|---|---|---|---|
| | | | zt w/ dis | zt w/o dis | st w/ dis | st w/o dis | mt | Overall | zt w/ dis | zt w/o dis | st w/ dis | st w/o dis | mt | Overall |
| w/o zt_head | 0.1 | 30.6 | 1.3 | 0.4 | 41.8 | 67.7 | 47.7 | 44.8 | 1.3 | 0.4 | 29.8 | 59.5 | 19.8 | 30.1 |
| w/o zt_head | 0.25 | 29.2 | 17.5 | 23.3 | 35.9 | 61.4 | 39.7 | 40.4 | 17.5 | 23.3 | 29.4 | 56.1 | 13.1 | 29.7 |
| w/o zt_head | 0.5 | 24.0 | 42.6 | 59.7 | 23.6 | 47.6 | 25.8 | 31.0 | 42.6 | 59.7 | 21.4 | 44.9 | 7.5 | 24.9 |
| w/ zt_head | 0.1 | 33.4 | 22.2 | 55.9 | 41.6 | 67.4 | 47.7 | 48.0 | 22.2 | 55.9 | 28.7 | 59.4 | 19.7 | 33.4 |
| w/ zt_head | 0.25 | 31.7 | 33.9 | 66.7 | 35.7 | 61.2 | 39.6 | 42.9 | 33.9 | 66.7 | 29.2 | 56 | 13.1 | 32.2 |
| w/ zt_head | 0.5 | 25.3 | 53.4 | 82 | 23.5 | 47.4 | 25.8 | 32.4 | 53.4 | 82 | 21.3 | 44.8 | 7.6 | 26.3 |

from the selected frames remain a challenge. As part of our future work, we plan to enhance the model's robustness to handle such cases effectively. Additionally, we aim to expand the dataset to include more challenging examples of this nature, which could serve as a new benchmark for advancing research in this area.

# D   GENERALIZED 3D REFERRING EXPRESSION SEGMENTATION

To validate the extensibility of our model, We also conducted experiments on the Generalized 3D Referring Expression Segmentation (3D-GRES) task (Wu et al., 2024a). Unlike the traditional 3D-RES task, the 3D-GRES task allows the number of targets to be arbitrary, including zero target, single target, and multiple targets. Due to the specificity of 3D-GRES, we designed an additional branch based on our Multi-Expert Network (MEN) to address the zero-target cases in 3D-GRES. For more details, see Sec. D.1.

To provide further clarification, all experiments presented in the main text are focused exclusively on the traditional 3D-RES task and do not consider the zero-target cases encountered in 3D-GRES task. All discussions regarding 3D-GRES are included in this section.

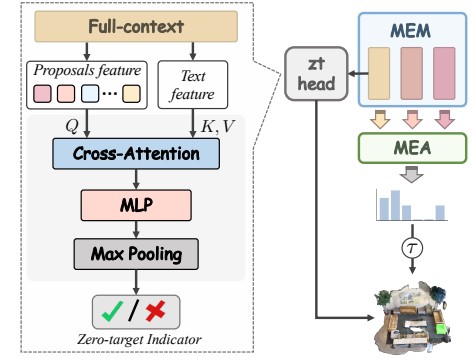

Figure 5: The structure of Zero-target head.

## D.1   ZERO-TARGET DETECTION HEAD

To handle the zero-target cases in 3D-GRES, we introduce a specifically designed Zero-Target Detection Head, as shown in Fig. 5. Specifically, the Zero-Target Detection Head decouples zero-target cases separately, serving as a parallel branch to the MEN, and independently predicts whether the current sample is a zero-target.

**Problem Definition.** The zero-target detection is treated as a classification task compatible with our weakly supervised setup. It uses only the category labels $Y_{zt} \in \mathbb{R}^2$, where $Y_{zt} = (0, 1)$ indicates "zero-target" and $Y_{zt} = (1, 0)$ indicates "not zero-target," without the need for costly point-level mask annotations.

The Zero-Target Detection Head takes as input the 2D full-context features and textual full-context features, both derived from the full-context output. Firstly, the cross-attention layer uses the textual feature $T_{full} \in \mathbb{R}^D$ as both the Value and Key input, and the visual proposal feature $F_{full} \in \mathbb{R}^{N_o \times D}$ as the Query input. This layer models the relationship between the textual description and each 3D proposal, and subsequently refines the Query:

$$Q_{full} = \text{SoftMax}(\frac{F_{full}W_q \cdot (T_{full}W_k)^T}{\sqrt{D}}) \cdot T_{full}W_v, \tag{D.1}$$

where $W_q, W_k, W_v \in \mathbb{D}^D$ are learnable parameters and $Q_{full} \in \mathbb{R}^{N_o \times D}$ denotes the refined Query. Following the vision-text interaction, $Q_{full}$ is processed through an MLP to generate zero-target prediction scores for each proposal. A MaxPooling layer is then applied to derive the zero-target label $E$:

$$E = \text{MaxPool}(\text{MLP}(Q_{full})), \tag{D.2}$$

where $\text{MaxPool}(\cdot)$ refer to the max pooling operation. The detection head is trained from scratch using a cross-entropy (CE) based classification loss $L_{zt}$:

$$L_{zt} = \text{CE}(E, Y_{zt}). \tag{D.3}$$

## D.2 INFERENCE

**Without zt_head.** During inference, for the original MEN model without the "zt_head", the masks of proposals with a probability mass $> 0.1$ are combined to form the predicted mask $M \in \{0, 1\}^{N_P}$. If there are no proposals with a probability mass greater than 0.1, the case is predicted as zero-target, and the predicted mask is set to all zeros.

**With zt_head.** Building on the threshold filtering described above, we introduce the "zt_head" to improve the model's ability to detect zero-target cases. Specifically, if the prediction from zt_head is "zero target," the predicted mask is set to all zeros. If the prediction is not "zero target," the mask is generated using the threshold filtering method described above.

## D.3 DATASET AND METRICS

**Multi3DRefer** (Zhang et al., 2023) includes 61,926 language expressions referring to 11,609 objects across 800 ScanNet scenes, with 6,688 expressions matching zero targets and 13,178 matching multiple targets. The dataset is originally introduced for 3D-REC. To accomplish the 3D-GRES, we utilized the benchmark provided by Wu et al. (2024a). We employing mIoU and "Acc@$m$IoU" as evaluation metrics, with $m \in \{0.25, 0.5\}$.

## D.4 RESULTS

**3D-GRES.** We present its results on the 3D-GRES, as shown in Tab.13. Without the zt_head, it can be observed that MEN achieves an mIoU of 30.6%. Notably, under the "st w/ dis" condition, MEN demonstrates a 0.3% improvement over the SOTAs in the Acc@0.5 metric. This highlights the strong discriminative power of our model and its remarkable potential for handling complex real-world scenarios. Additionally, it is important to highlight that the inclusion of the designed zt_head significantly improves our model's ability to handle the specific "zero-target" cases. The introduction of the zt_head yields improvements of 20.9% and 55.5% in Acc@0.5 for the "zt w/ dis" and "zt w/o dis" conditions, respectively. At the same time, the accuracy for other scenarios, such as single and multi-target, does not decrease. This is because the zt_head rarely misclassifies a target as zero-target, so it does not reduce the recall for single and multi-target cases, thus maintaining robustness for those.

**Ablation Results on Thresholds and Zero-Target Detection Head.** We analyzed the impact of different confidence thresholds on the performance of our MEN in 3D-GRES, as shown in Tab. 14. It can be observed that when the threshold is set low, such as at 0.1, it performs well in the presence of target objects but shows suboptimal results in the "zt" subset. In contrast, when the threshold is set higher, such as at 0.5, a large number of samples will be incorrectly predicted as "zt". While performance on "zt" improves, this is achieved at the expense of a significant decline in performance

on other subsets. This is because the semantic distribution output by the MEN is relative to the objects in the entire scene and does not correspond to the absolute matching probability between the 3D proposal and the input text. Therefore, for our model, directly applying a threshold to filter instances is not suitable for the 3D-GRES task.

The impact of different thresholds exhibits a similar pattern following the introduction of the zt_head. However, it is important to note that, with the assistance of the zt_head, the model's performance on the "zt" subset shows significant improvement across all thresholds. We can also observe that the introduction of the zt_head does not result in a decrease in accuracy for single and multi-target cases. This further highlights the zt_head's robustness across these scenarios. Therefore, it is necessary to design a separate prediction branch specifically for the special zero-target case.

## E    CALCULATION OF VISIBILITY

To enhance the clarity of the paper, we refer to the visibility calculation process in Boudjoghra et al. (2024), as follows:

Given the $N_f$ 2D-RGB frames $\mathcal{I} = \{\mathcal{I}_j \in \mathbb{R}^{4 \times W \times H} | \forall j \in (1, \ldots, N_f)\}$ associated with the 3D point cloud $P$, along with their intrinsic matrices $I \in \mathbb{R}^{N_f \times 4 \times 4}$ and extrinsic matrices $E \in \mathbb{R}^{N_f \times 4 \times 4}$. Due to the projection of the 3D point cloud $P$ on a frame with index $i$ can be computed as follows $P_i^{2D} = I_i \cdot E_i \cdot P^{3D}$, the projection of the point cloud scene $P$ onto all frames can be computed in parallel as follows:

$$P^{2D} = (I \star E) \cdot P^{3D}, \tag{E.1}$$

where $\star$ is batch matrix multiplication, $P^{3D} \in \mathbb{R}^{4 \times N_p}$ represents the point cloud in homogeneous coordinates, and $P^{2D} \in \mathbb{R}^{N_f \times 4 \times N_p}$ denotes the corresponding 2D homogeneous coordinates. Next, we compute the visibility $V^c, V^d \in \mathbb{R}^{N_f \times N_p}$ of points in both the 2D coordinate space and the depth space, as follows:

$$V^c = \mathbf{1}_{\text{condition}}(0 < P_x^{2d} < W) \odot \mathbf{1}_{\text{condition}}(0 < P_y^{2d} < H), \tag{E.2}$$

$$V^d = \mathbf{1}_{\text{condition}}(|P_d^{2d} - D_z| < \tau_{depth}), \tag{E.3}$$

where $\mathbf{1}_{\text{condition}}$ denotes the indicator function, $\odot$ represents element-wise multiplication, $W$ and $H$ refer to the image width and height respectively, $D_z \in \mathbb{R}^{N_f \times N_p}$ is the true depth of the 3D points obtained from the depth maps, $|\cdot|$ indicates the operation of taking the absolute value, and $P_x^{2D}, P_y^{2D}, P_z^{2D} \in \mathbb{R}^{N_f \times N_p}$ represent the x and y coordinates and depth of the 3D projected points across all frames, respectively. Then, the per 3D mask proposal visibility $V \in \mathbb{R}^{N_o \times N_f}$ can be computed as follows:

$$V = \mathcal{M} \cdot (V^c \odot V^d)^T, \tag{E.4}$$

where $\mathcal{M} \in \mathbb{R}^{N_o \times N_p}$ denotes the matrix of 3D mask proposals. The total number of visible points of the 3D proposal $i$ in frame $j$ is denoted as $V_{i,j} \in V$.

## F    THE THEORETICAL PROOF OF THEOREM 3.1.

To improve the clarity of our paper, we refer to the proof of Theorem 3.1 in Hill (2011), as follows:

### F.1    BASIC DEFINITION AND PROPERTIES OF CONFLATIONS

Throughout this article, $\mathbb{N}$ will denote the natural numbers, $\mathbb{Z}$ the integers, $\mathbb{R}$ the real numbers, $(a, b]$ the half-open interval $\{x \in \mathbb{R} : a < x \leq b\}$, $\mathbb{B}$ the Borel subsets of $\mathbb{R}$, $\mathcal{P}$ the set of all real Borel probability measures, $\delta_x$ the Dirac delta measure in $\mathcal{P}$ at the point $x$ (i.e., $\delta_x(B) = 1$ if $x \in B$, and $= 0$ if $x \notin B$), $\|\mu\|$ the total mass of the Borel sub-probability $\mu$, $o(\ )$ the standard "little oh" notation $o(a_n) = b_n$ if and only if $\lim_{n \to \infty} \frac{a_n}{b_n} = 0$, *a.c.* means absolutely continuous, the *p.m.f.* of $P$ is the probability mass function $(p(k) = P(\{k\}))$ if $P$ is discrete and *p.d.f.* is the probability density function (Radon-Nikodyn derivative) of $P$ if $P$ is a.c., $E(X)$ denotes the expected value of the random variable $X$, $\psi_P$ the characteristic function of $P \in \mathcal{P}$ (i.e., $\psi_P(t) = \int_{-\infty}^{\infty} e^{itx} dP(x))$, $I_A$ is the indicator function of the set $A$ (i.e. $I_A(x) = 1$ if $x \in A$ and $= 0$ if $x \notin A$), $g \otimes h$ is the

convolution $(g \otimes h)(t) = \int_{-\infty}^{\infty} g(t-s)h(s)ds$ of $g$ and $h$, and $A^c$ is the complement $\mathbb{R} \backslash A$ of the set $A$. For brevity, $\mu((a,b])$ will be written $\mu(a,b]$, $\mu(\{x\})$ as $\mu(x)$, etc.

**Definition F.1.** For $P_1, \ldots, P_n \in \mathcal{P}$ and $j \in \mathbb{N}$, $\mu_j(P_1, \ldots, P_n)$ is the purely-atomic $j$-dyadic sub-probability measure

$$\mu_j(P_1, \ldots, P_n) = \sum_{k \in \mathbb{Z}} \prod_{i=1}^{n} P_i((k-1)2^{-j}, k2^{-j}] \delta_{k2^{-j}}.$$

**Remark.** The choice of using half-open dyadic intervals closed on the right, and of placing the mass in every dyadic interval at the right end point is not at all important — the results which follow also hold if other conventions are used, such as decimal or ternary half-open intervals closed on the left, with masses placed at the center.

**Example F.2.** If $P_1$ is a Bernoulli distribution with parameter $p = \frac{1}{3}$ (i.e. $P = \frac{(2\delta_0 + \delta_1)}{3}$) and $P_2$ is Bernoulli with parameter $\frac{1}{4}$, then $\mu_j(P_1, P_2) = \frac{(6\delta_{1/2} + \delta_1)}{12}$ for all $j \in \mathbb{N}$.

The next proposition is the basis for the definition of conflation of general distributions below. Recall that for real Borel sub-probability measures $\{\nu_j\}$ and $\nu$, the following are equivalent:

$$\nu_j \to \nu \text{ vaguely as } j \to \infty; \tag{F.1a}$$

$$\nu_j(a,b] \to \nu(a,b] \text{ for all } a < b \text{ in a dense set } D \subset \mathbb{R}; \tag{F.1b}$$

$$\lim_{j \to \infty} \int f(x)d\nu_j(x) = \int f(x)d\nu(x) \tag{F.1c}$$

for all continuous $f$ that vanish at infinity.

**Theorem F.3.** *For all* $P, P_1, \ldots P_n \in \mathcal{P}$

(i) $\mu_{j+1}\left(\frac{a}{2^m}, \frac{b}{2^m}\right] \leq \mu_j\left(\frac{a}{2^m}, \frac{b}{2^m}\right]$ *for all* $j, m \in \mathbb{N}$, $j > m$; *and all* $a \leq b$, $a, b \in \mathbb{Z}$;

(ii) $\mu_j(P_1, \ldots, P_n)$ *converges vaguely to a sub-probability measure*

$\mu_\infty(P_1, \ldots, P_n)$;

(iii) $\lim_{j \to \infty} \|\mu_j(P_1, \ldots, P_n)\| = \|\mu_\infty(P_1, \ldots, P_n)\|$; *and*

(iv) $\mu_\infty(P) = P$, *and* $\mu_j(P)$ *converges vaguely to* $P$ *as* $j \to \infty$.

The following simple observation — that the square of the sums of nonnegative numbers is always at least as large as the sum of the squares — will be used in the proof of the theorem and several times in the sequel, and is recorded here for ease of reference.

**Lemma 1.** *For all* $n \in \mathbb{N}$, *all* $a_{i,k} \geq 0$, *and all* $J \subset \mathbb{N}$, $\prod_{i=1}^{n} \sum_{k \in J} a_{i,k} \geq \sum_{k \in J} \prod_{i=1}^{n} a_{i,k}$.

## F.2 CONFLATIONS OF DISCRETE DISTRIBUTIONS

*Proof.* Fix $P_1, \ldots, P_n$ and note that by definition of atom, $p_i(x) > 0$ for all $i = 1, \ldots, n$ and all $x \in A$. Fix $k_0 \in \mathbb{Z}$ and $j_0 \in \mathbb{N}$, and let $D = \left(\frac{k_0}{2^{j_0}}, \frac{k_0+1}{2^{j_0}}\right]$. First it will be shown that

$$\mu_\infty(D) = \sum_{x \in A \cap D} \prod_{i=1}^{n} p_i(x). \tag{F.2}$$

For all $x \in \mathbb{R}$, $j \in \mathbb{N}$, let $D_{x,j}$ denote the unique dyadic interval $\left(\frac{k}{2^j}, \frac{k+1}{2^j}\right]$ containing $x$. Note that $D_{x,j} \searrow \{x\}$ as $j \to \infty$ so $P_i(D_{x,j}) \searrow p_i(x)$ as $j \to \infty$ for all $i$ and all $x \in \mathbb{R}$.

This implies

$$\lim_{j \to \infty} \prod_{i=1}^{n} P_i(D_{x,j}) = \prod_{i=1}^{n} p_i(x) \quad \text{for all } x \in \mathbb{R}. \tag{F.3}$$

Fix $\epsilon > 0$. Since $\{P_i\}$ are discrete, there exists a finite set $A_0 \subset \mathbb{R}$ such that

$$P_i(D \cap A_0^c) < \epsilon \quad \text{for all } i \in \{1, \ldots, n\}. \tag{F.4}$$

Since $\prod_{i=1}^{n} p_i(x) = 0$ for all $x \in A^c$, equation F.4 implies

$$\Big| \sum_{x \in A \cap D} \prod_{i=1}^{n} p_i(x) - \sum_{x \in A_0 \cap D} \prod_{i=1}^{n} p_i(x) \Big| = \sum_{x \in A \cap A_0^c \cap D} \prod_{i=1}^{n} p_i(x) \tag{F.5}$$

$$\leq \sum_{x \in A \cap A_0^c \cap D} p_1(x) \leq P_1(D \cap A_0^c) < \epsilon.$$

For each $j \in \mathbb{N}$, let $S_j = \bigcup_{x \in A_0} D_{x,j}$. Then since $x \in D_{x,j}$ for all $x$ and $j$, equation F.4 implies $P_i(D \cap S_j^c) < \epsilon$ for all $i \in \{1, \ldots, n\}$. Thus by definition of $\{\mu_j\}$ and Lemma 1,

$$\mu_j(D \cap S_j^c) \leq \prod_{i=1}^{n} P_i(D \cap S_j^c) < \epsilon^n \qquad \text{for all } j \in \mathbb{N}. \tag{F.6}$$

This implies that

$$\mu_j(D) = \mu_j(D \cap S_j) + \mu_j(D \cap S_j^c) \tag{F.7}$$

$$= \sum_{x \in D \cap A_0} \mu_j(D_{x,j}) + \mu_j(D \cap S_j^c)$$

$$= \sum_{x \in D \cap A_0} \prod_{i=1}^{n} P_i(D_{x,j}) + \mu_j(D \cap S_j^c)$$

where the second equality follows from the definitions of $S_j$ and $D_{x,j}$. Since $x \in D_{x,j}$, equation F.7 implies

$$\mu_j(D) \geq \sum_{x \in D \cap A_0} \prod_{i=1}^{n} P_i(D_{x,j}) \geq \sum_{x \in D \cap A_0} \prod_{i=1}^{n} p_i(x). \tag{F.8}$$

By equation F.7, equation F.3 and equation F.6,

$$\mu_j(D) \leq \sum_{x \in A_0 \cap D} \prod_{i=1}^{n} p_i(x) + \epsilon^n + \epsilon \quad \text{for sufficiently large } j. \tag{F.9}$$

By equation F.8 and equation F.9, $|\mu_j(D) = \sum_{x \in A_0 \cap D} \prod_{i=1}^{n} p_i(x)| \leq \epsilon + \epsilon^n$, so by equation F.5, $|\mu_j(D) - \sum_{x \in A \cap D} \prod_{i=1}^{n} p_i(x)| < 2\epsilon + \epsilon^n$. Since $\epsilon > 0$ was arbitrary and since $\mu_j \to \mu_\infty$, this implies equation F.2. Since $D$ was arbitrary, equation F.2 implies that $\|\mu_\infty(P_1, \ldots, P_n)\| = \sum_{x \in A} \prod_{i=1}^{n} p_i(x)$, which proves that $\&(P_1, \ldots, P_n)$ exists. The equivalence of (i) and (ii) follows since $\&(P_1, \ldots, P_n) = \frac{\mu_\infty}{\|\mu_\infty\|}$ and since the measures of dyadic intervals $D$ determine $\mu_\infty$. The equivalence of (ii) and (iii) follows immediately from the definition of conditional probability. $\qquad \square$

