# OpenReview forum: "Weakly-supervised 3D Referring Expression Segmentation"
_ICLR.cc/2025/Conference — Submitted to ICLR 2025_

### Official Review · Reviewer_nJzX · 2024-10-27

**Soundness:** 2
**Presentation:** 2
**Contribution:** 2
**Rating:** 3
**Confidence:** 5

**Summary:**

This paper proposes a new weakly supervised 3D Referring Expression Segmentation task, along with a baseline method, named Multi-Expert Network (MEN), designed to address this task. The proposed task can potentially eliminate the need for laborious point-level mask annotations. The proposed method includes two main components: a Multi-Expert Mining module, which leverages various experts to extract and align visual and textual features from multiple complementary cues, and a Multi-Expert Aggregation module, which integrates the knowledge acquired from these experts.

**Strengths:**

1. The proposed 3D-RES task is well motivated.
2. The proposed MEN method is reasonable and all proposed modules are thoroughly ablated.

**Weaknesses:**

1. The proposed method employs SoftGroup as its instance segmentation backbone. However, since this model was trained on ScanNet data, there is a significant data leakage issue, as both evaluation datasets (ScanRefer and Multi3DRefer) were built using ScanNet. Despite this favorable evaluation setting, the method's performance remains substantially lower than fully supervised approaches. The performance would likely deteriorate further when applied to other datasets. Given that such a method is primarily intended for use in unseen environments, the poor generalization capability presents a major concern and undermines the core motivation of this work. An ablation using an alternative backbone trained on different data may help to alleviate this concern.
2. Based on Table 1, distilling MEN to fully-supervised methods does not improve performance in the "Multiple" and "Overall" settings. This outcome suggests that the segmentation masks obtained through this method are noisy and struggle to differentiate between multiple objects, rendering the supervision signal less effective. Table 2 further supports this observation, showing poor performance in both zero-target and multiple-target cases. These limitations imply that the model may lack the ability to accurately interpret referring expressions to distinguish between multiple objects. This raises the question of whether there is any difference between using referring expressions and class labels for training this model. An ablation study on this would help to clear the confusion.
3. The proposed method, while a commendable effort by the author, lacks elegance. It primarily represents a complex engineering solution that combines various existing approaches and models.
4. Typo:
> 1. Line 201: Dimension of RGB-D should not be 3xWxH.
> 2. Line 255: textaul -> textual.
5. Paper has organizational issues. Many core details are not mentioned or put in the appendix.

**Questions:**

Refer to weaknesses.

---

> ### Author Response · Authors · 2024-11-24
> **Official Comment by Authors (part-1)**
>
> > **W1: The proposed method employs SoftGroup as its instance segmentation backbone. However, since this model was trained on ScanNet data, there is a significant data leakage issue, as both evaluation datasets (ScanRefer and Multi3DRefer) were built using ScanNet. Despite this favorable evaluation setting, the method's performance remains substantially lower than fully supervised approaches. The performance would likely deteriorate further when applied to other datasets. Given that such a method is primarily intended for use in unseen environments, the poor generalization capability presents a major concern and undermines the core motivation of this work. An ablation using an alternative backbone trained on different data may help to alleviate this concern.**
> >
>
> A1: Thank you for your thoughtful suggestion. There is no data leakage in our method, as the SoftGroup model used as the backbone was trained on the ScanNet training set, which does not overlap with the evaluation datasets in ScanRefer and Multi3DRefer.
>
> To address your concern, we added an additional experiment where we replaced the SoftGroup backbone with a model trained on the S3DIS dataset [B]. The results, included in the updated table below, demonstrate that our model maintains its robustness, achieving a consistent improvement of 13.9% in mIoU over the baseline, even with this alternative backbone.
>
> These findings confirm that our method generalizes effectively and is not overly reliant on the original training data of the SoftGroup backbone. We have added this discussion to Section A.3 in the updated version. We sincerely appreciate your suggestion, as it has helped further validate the flexibility and robustness of our approach.
>
> Table a: 3D-RES results on ScanRefer, where the instance segmentation models for both the baseline and our MEN are pre-trained on the S3DIS dataset, but evaluated on the ScanNet scenes.
>
> |  | mIoU | Unique Acc\@0.25 | Unique Acc\@0.5 | Multiple Acc\@0.25 | Multiple Acc\@0.5 | Overall Acc\@0.25 | Overall Acc\@0.5 |
> | --- | --- | --- | --- | --- | --- | --- | --- |
> | Baseline | 10.6 | 33.1 | 28.6 | 9.8 | 6.9 | 14.3 | 11.1 |
> | Ours | 24.5 | 59.7 | 52.1 | 27.4 | 19.9 | 33.7 | 26.2 |
>
> [A] Softgroup for 3d instance segmentation on point clouds. In CVPR 2022.
>
> [B] STPLS3D: A Large-Scale Synthetic and Real Aerial Photogrammetry 3D Point Cloud Dataset. In BMVC 2022.
>
> > **W2-1: Based on Table 1, distilling MEN to fully-supervised methods does not improve performance in the "Multiple" and "Overall" settings. This outcome suggests that the segmentation masks obtained through this method are noisy and struggle to differentiate between multiple objects, rendering the supervision signal less effective.**
> >
>
> A2-1: Thank you for highlighting this observation. The "Multiple" setting is inherently complex and poses substantial challenges. While our pseudo-labels are of significantly higher quality than the baseline, their accuracy in this setting is still below 40%. Consequently, over 60% of the supervisory signals in the fully supervised model are noisy or erroneous, which limits the potential performance improvement in this scenario.
>
> However, this does not indicate that our model struggles to handle the "Multiple" setting. In fact, with the integration of our carefully designed MEM module, the model achieves a remarkable improvement of 13.9% in "R\@1, IoU\@0.5" over the baseline in the "Multiple" setting. This highlights the robustness and effectiveness of our approach in tackling these challenging cases, even under noisy supervision conditions.
>
> > **W2-2: Table 2 further supports this observation, showing poor performance in zero-target cases.**
> >
>
> A2-2: For the 3D-GRES task, both our model and the baseline lack a carefully designed approach for handling zero-target.  As mentioned in Lines 363–365, our current pipeline combines the masks of proposals with probabilities greater than 0.1 to form the predicted mask. If no proposals exceed this threshold, the case is predicted as zero-target. To evaluate this mechanism, we analyzed the impact of varying thresholds on 3D-GRES performance, as presented in Table 10 of the supplementary materials. While increasing the threshold improved zero-target (zt) detection, it negatively affected performance on multi-target (mt) cases, showing that threshold adjustments alone cannot effectively address the zero-target problem.
>
> Inspired by your feedback, we explored an additional approach to improve zero-target detection without compromising performance in other cases. Specifically, we introduced a dedicated *zt_head*, treating zero-target detection as a classification task compatible with our weakly supervised setup. It uses only category labels without adding expensive mask labels, as explained in section D of the updated paper.

---

> ### Author Response · Authors · 2024-11-24
> **Official Comment by Authors (part-2)**
>
> **Supplement to A2-2:**
>
> The results demonstrate that the *zt_head* significantly improved performance on zero-target cases, without negatively impacting other scenarios. This enhancement highlights the robustness of the proposed pipeline.
>
> We greatly appreciate your valuable suggestion, and we have incorporated this adjustment into  the next version (highlighted in blue, lines 365–367, 429-431, 945-977). This enhancement not only enriches the content of the paper but also improves the robustness of the model. Thank you again for your thoughtful input.
>
> Table b: Results of 3D-GRES, including mIoU and Acc\@0.5, where "w/o zt_head" means that the zt_head was not used, and "w/ zt_head" means that the zt_head was used to predict whether it is a zero target.
>
> |  | miou | zt  w/  dis | zt  w/o  dis | st  w/ dis | st  w/o dis | mt | overall |
> | --- | --- | --- | --- | --- | --- | --- | --- |
> | w/o zt_head | 30.6 | 1.3 | 0.4 | 28.9 | 59.5 | 19.8 | 30.1 |
> | w/ zt_head | 33.4 | 22.2 | 55.9 | 28.7 | 59.4 | 19.7 | 33.4 |
>
> > **W2-3: This raises the question of whether there is any difference between using referring expressions and class labels for training this model. An ablation study on this would help to clear the confusion.**
> >
>
> A2-3: Thank you for your question. Our experiments, detailed in Table 3, include ablation studies comparing the use of referring expressions and class labels for training.
>
> - **Class Labels Only (Row 3):** Using only class labels proves insufficient for the "Multiple" setting, achieving an Acc@0.5 of just 17.4%, as it lacks the specificity needed to resolve ambiguities when multiple proposals are present.
> - **Referring Expressions Only (Row 1):** Using only referring expressions improves performance in the "Multiple" setting. However, when the selected RGB-D frames contain multiple proposals, the lack of category-specific grounding can lead to significant matching ambiguity.
> - **Combination (Row 5):** Combining referring expressions and class labels increases Acc@0.5 in the "Multiple" setting to 27.1%, highlighting the complementary strengths of both signals.
> - **Attribute Experts (Row 7):** Incorporating attribute experts, which extract target-specific attribute information from referring expressions, further boosts localization accuracy, resulting in an additional 0.8% improvement in the "Multiple" setting.
>
> These results demonstrate that the combination of referring expressions, class labels, and attribute experts is critical for effectively resolving ambiguities and improving performance in challenging scenarios like the "Multiple" setting. Thank you again for your valuable observation.
>
> > **W3: The proposed method, while a commendable effort by the author, lacks elegance. It primarily represents a complex engineering solution that combines various existing approaches and models.**
> >
>
> A3: Thank you for your feedback. We respectfully disagree with the characterization of our work as merely a complex engineering solution. The contributions of this paper are substantial and address critical challenges in the field, as outlined below:
>
> - **Core Contributions**
> 1. **New Task Setting:**
>
>     We propose the *weakly-supervised 3D-RES task*, reducing reliance on costly mask-level annotations and paving the way for efficient multimodal research.
>
> 2. **MEN Framework:**
>
>     To address the absence of image-text pair masks in this new task, we developed the MEN framework. This framework introduces multiple experts to extract complementary semantic cues. We also propose a novel, parameter-free *MoE* approach, MEA,  validated through mathematical derivation, offering a lightweight yet effective solution for expert modeling. This establishes a strong and versatile baseline for future research.
>
> - Additional Insights
>
> In addition to these core contributions, we present valuable advancements that further enhance the robustness of our approach and provide insights for future research:
>
> 1. **Full-Context Expert:**
>
>     Addressing the limitations of standard CLIP models in handling complex, long textual descriptions [A], we developed the full-context expert, which uses Long-CLIP for richer text encoding and integrates complete visual context. This improves multimodal matching, as demonstrated by:
>
>     - A 3.7% mIoU gain when replacing CLIP with Long-CLIP (Table 6, rows 2 and 3).
>     - A 9.3% mIoU improvement when incorporating complete visual context instead of rendered images (Table 5, rows 1 and 4).
> 2. **Attribute and Class Experts:**
>
>     These experts provide complementary semantic cues, significantly enhancing performance by resolving ambiguities, as shown in Table 3.

---

> ### Author Response · Authors · 2024-11-24
> **Official Comment by Authors (part-3)**
>
> **Supplement to A3**
>
> The paper's contributions are both foundational and impactful, addressing critical challenges with novel solutions that extend beyond simple engineering. We believe these advancements significantly enrich the field, offering practical tools and insights for future research. Thank you again for your feedback, which has helped us further emphasize the significance of our work.
>
> [A] Long-CLIP: Unlocking the Long-Text Capability of CLIP. In ECCV 2025.
>
> > **W4: Typo：**
> >
> >
> > > **1. Line 201: Dimension of RGB-D should not be 3xWxH.**
> > >
> > > **2. Line 255: textaul -> textual.**
>
> A4:  Thank you for your careful attention to detail. We have made the necessary corrections to the errors in Line 201 and Line 255 (highlighted in blue) in the updated version of the paper.

---

> ### Comment · Reviewer_nJzX · 2024-11-25
>
> Thank you very much for the author's response. I have made every effort to thoroughly read all the content. However, overall, I find myself feeling even more confused after reading the response, as it seems to raise additional inconsistencies.
>
> **A1:** The author claims that the method's backbone is not reliant on ScanNet training by retraining their methods on S3DIS. However, it seems like the S3DIS results of 24.5 and the ScanNet results of 33.4 on Tab. 1  have a huge gap, which by all means signifiy to me it is indeed sensitive to backbone training data.
>
> **A2-1:** The author mentions that 1) "over 60% of the supervisory signals in the fully supervised model are noisy or erroneous" and 2) the distillation results in Tab. 1 shows improvement in Unique case but degragation in Multiple case. Based on these two pieces of information, I can only think of one possibility. The noise is mostly in Multiple case, which aligns with my observation that the supervision signal is not effective at differentiating between multiple objects.
>
> **A2-2:** I am glad to see the author resolving the low zero-target problem by proposing the use of a separate zt_head to greatly improve its performance in this scenario. However, currently the author just abruptly inserts this new setting into the experiment setting section and makes the whole paper very unorganized. It should be carefully reorganized (e.g. maybe it mentioned as part of the method and ablated the choices in the experimental sections).
>
> **A2-3:** I think the author is misunderstanding my question. I am mainly curious how much advantage does the proposed weakly supervised methods have over fully supervised learning methods using class labels. Therefore, I want to see comparison with leading supervised learning methods  using class labels, not for additional ablations on the author own proposed modules.
>
> **A3:** While the proposed new task can in theory reduce the costly mask-level annotations, the presented experiments does not seem to support it as we observe substantial drop in accuracy compared to methods with annotations. With regards to the complexity of the method, as I mention, I am not discrediting the author effort. The engineering effort is indeed significant, with the proposal of MEN that consists of multiple intricately designed experts. But elegance does not equate to complexity. Not much new insight can be derived, aside from knowing that piecing multiple experts can give us good results, which in some ways are expected.
>
> Additional questions:
> 1. Upon checking Tab. 1 multiple times, I am curious why R@3 results are missing for Ours(STMN) and Ours (MDIN)?
> 2. I agree with reviewer kMXL W1's comment and believe that the response provided does not adequately address the issue. The claim that it is relatively rare in ScanRefer is not a convincing argument. If anything, it suggests that ScanRefer may not be a suitable dataset for this purpose, further undermining the credibility of the experimental results presented.
> 3. The additional results provided for Open-Vocabulary Methods appear unusually low. Upon reviewing the results in the cited papers, it seems that they primarily focus on ScanNet 200. To facilitate comparison, it may be helpful to run MEN on this dataset and reference the results from their paper. Since the author claims in A1 that the model is not sensitive to the training data, no retraining should be necessary and only inference should be required.

---

> ### Author Response · Authors · 2024-11-25
> **Response to Reviewer nJzX (part-1)**
>
> Thank you for your prompt and thoughtful feedback, as well as your recognition of our contributions. We greatly appreciate the effort you have invested in evaluating our work. Your insightful suggestions have been highly inspiring and have helped us improve our approach. Below, we provide a more detailed response to address your concerns and further clarify our methodology.
>
> > **A1:** The author claims that the method's backbone is not reliant on ScanNet training by retraining their methods on S3DIS. However, it seems like the S3DIS results of 24.5 and the ScanNet results of 33.4 on Tab. 1 have a huge gap, which by all means signifiy to me it is indeed sensitive to backbone training data.
> >
>
> R1: Thank you for your timely response. First, we would like to clarify that we did not claim the method’s backbone is not reliant on ScanNet training. Rather, we emphasized that the superiority of our method over the baseline does not depend on the specific training data used for the backbone.
>
> In our previous response, we followed your suggestion and trained the backbone using the S3DIS dataset. We then evaluated both our method and the baseline on the ScanNet test set using this new configuration. The results show that our method still achieves significant improvement over the baseline, with a performance gain of 13.9. It is important to note that this setup involves a cross-domain issue between the training and testing datasets, which explains why the observed performance of 24.5 is lower than the 33.4 reported in the original paper, where both the backbone training and final testing were conducted on ScanNet. This difference is expected due to the domain shift [1,2,3].
>
> [1] Cross-Domain Semantic Segmentation via Domain-Invariant Interactive Relation Transfer. In CVPR 2020.
>
> [2] Train in Germany, Test in The USA: Making 3D Object Detectors Generalize. In CVPR 2020.
>
> [3] Undoing the Damage of Label Shift for Cross-domain Semantic Segmentation. In CVPR 2022.
>
> > **A2-1:** The author mentions that 1) "over 60% of the supervisory signals in the fully supervised model are noisy or erroneous" and 2) the distillation results in Tab. 1 shows improvement in Unique case but degragation in Multiple case. Based on these two pieces of information, I can only think of one possibility. The noise is mostly in Multiple case, which aligns with my observation that the supervision signal is not effective at differentiating between multiple objects.
> >
>
> R2-1: The "Multiple" case is inherently a challenging problem, and it is intuitive that more noise would be present in such cases. This observation is consistent with previous studies [4-16], all of which have found that performance on multi-object cases tends to be worse than on single-object cases. Therefore, this is not a unique issue to our work and should not be considered a limitation.
>
> [4] ScanRefer: 3D Object Localization in RGB-D Scans using Natural Language. In ECCV 2020.
>
> [5] Free-form Description Guided 3D Visual Graph Network for Object Grounding in Point Cloud. In ICCV 2021.
>
> [6] SAT: 2D Semantics Assisted Training for 3D Visual Grounding. In ICCV 2021.
>
> [7] 3DVG-Transformer: Relation Modeling for Visual Grounding on Point Clouds. In ICCV 2021.
>
> [8] Text-Guided Graph Neural Networks for Referring 3D Instance Segmentation. In AAAI 2021.
>
> [9] 3D-SPS: Single-Stage 3D Visual Grounding via Referred Point Progressive Selection. In CVPR 2022.
>
> [10] 3DJCG: A Unified Framework for Joint Dense Captioning and Visual Grounding on 3D Point Clouds. In CVPR 2022.
>
> [11] ViewRefer: Grasp the Multi-view Knowledge for 3D Visual Grounding. In ICCV 2023.
>
> [12] EDA: Explicit Text-Decoupling and Dense Alignment for 3D Visual Grounding. In CVPR 2023.
>
> [13] X-RefSeg3D: Enhancing Referring 3D Instance Segmentation via Structured Cross-Modal Graph Neural Networks. In AAAI 2024.
>
> [14] 3D-STMN: Dependency-Driven Superpoint-Text Matching Network for End-to-End 3D Referring Expression Segmentation. In AAAI 2024.
>
> [15] SegPoint: Segment Any Point Cloud via Large Language Model. In ECCV 2024.
>
> [16] RefMask3D: Language-Guided Transformer for 3D Referring Segmentation. In ACM MM 2024.
>
> > **A2-2:** I am glad to see the author resolving the low zero-target problem by proposing the use of a separate zt_head to greatly improve its performance in this scenario. However, currently the author just abruptly inserts this new setting into the experiment setting section and makes the whole paper very unorganized. It should be carefully reorganized (e.g. maybe it mentioned as part of the method and ablated the choices in the experimental sections).
> >
>
> R2-2: Thank you for your valuable suggestions. We will reorganize the paper as per your feedback and continue optimizing it.

---

> ### Author Response · Authors · 2024-11-25
> **Response to Reviewer nJzX (part-2)**
>
> > **A2-3:** I think the author is misunderstanding my question. I am mainly curious how much advantage does the proposed weakly supervised methods have over fully supervised learning methods using class labels. Therefore, I want to see comparison with leading supervised learning methods using class labels, not for additional ablations on the author own proposed modules.
> >
>
> R2-3: Thank you for your thoughtful response. Using class labels for supervision is inherently a weakly supervised setting for 3D-RES, similar to the approach used in Weakly Supervised 3D-REC [17]. Therefore, there are no fully supervised learning methods that rely on class labels in this context.
>
> [17] Distilling Coarse-to-Fine Semantic Matching Knowledge for Weakly Supervised 3D Visual Grounding. In ICCV 2023.
>
> > **A3:** While the proposed new task can in theory reduce the costly mask-level annotations, the presented experiments does not seem to support it as we observe substantial drop in accuracy compared to methods with annotations. With regards to the complexity of the method, as I mention, I am not discrediting the author effort. The engineering effort is indeed significant, with the proposal of MEN that consists of multiple intricately designed experts. But elegance does not equate to complexity. Not much new insight can be derived, aside from knowing that piecing multiple experts can give us good results, which in some ways are expected.
> >
>
> R3: Thank you for recognizing the significance of the task setting and our engineering contributions. As the first work in this field, achieving ultra-high performance is not the primary goal. The value lies in introducing a new task and uncovering the challenges inherent in that task.
>
> To achieve this, we have thoroughly analyzed the challenges associated with the task, exploring it from multiple angles and extracting as much relevant information as possible from the visual-textual cues. Our aim is to provide meaningful insights for future researchers. This is why we included a detailed analysis in the experimental section, explaining what each expert addresses, as shown in Section 4.6. Through this process, we naturally developed a strong baseline.
>
> In summary, by introducing a new setting, showcasing its challenges, providing sufficient analysis, and offering a robust baseline, we believe this work represents a comprehensive and practical form of "elegance" that goes far beyond a pure engineering solution.
>
> > Additional questions-1: Upon checking Tab. 1 multiple times, I am curious why R@3 results are missing for Ours(STMN) and Ours (MDIN)?
> >
>
> R4:  Thank you for your attention to detail. In fact, both Ours (STMN) and Ours (MDIN) are single-stage models, and the R@3 metric is not applicable here. The R@3 metric refers to selecting the proposal with the highest IoU from the top-3 proposals [17]. However, in single-stage models, only a single proposal is output, making the R@3 metric not suitable for these models.
>
> [17] Distilling Coarse-to-Fine Semantic Matching Knowledge for Weakly Supervised 3D Visual Grounding. In ICCV 2023.
>
> > Additional questions-2: I agree with reviewer kMXL W1's comment and believe that the response provided does not adequately address the issue. The claim that it is relatively rare in ScanRefer is not a convincing argument. If anything, it suggests that ScanRefer may not be a suitable dataset for this purpose, further undermining the credibility of the experimental results presented.
> >
>
> R5: We appreciate the opportunity to engage with you on this issue. We fully acknowledge the validity of the scenario raised by reviewer kMXL and admit that our method does not specifically address this phenomenon. However, it is important to note that no benchmark or setting can cover every possible real-world scenario, and all models inevitably have limitations.
>
> For instance, early benchmarks like CIFAR-10 [18] were designed for a simple 10-class classification task, and could not handle the complexity of larger datasets like ImageNet, which contains 1,000 categories. Similarly, early Visual Question Answering (VQA) datasets were biased and allowed models to infer answers solely based on the text, without considering the image [19]. Over time, these datasets were improved to better reflect real-world challenges, but this does not diminish the significance of their initial contributions.
>
> Research, much like the construction of Rome, is built gradually. The scenario raised by reviewer kMXL is indeed an interesting and valuable direction for future work. We and subsequent researchers will continue to make strides in this area. However, this does not undermine the contributions of our paper or "undermine the credibility of the experimental results presented."
>
> [18] Learning multiple layers of features from tiny images. Krizhevsky et al. 2009
>
> [19] Making the V in VQA Matter: Elevating the Role of Image Understanding in Visual Question Answering. In CVPR 2017.

---

> ### Author Response · Authors · 2024-11-25
> **Response to Reviewer nJzX (part-3)**
>
> > Additional questions-3: The additional results provided for Open-Vocabulary Methods appear unusually low. Upon reviewing the results in the cited papers, it seems that they primarily focus on ScanNet 200. To facilitate comparison, it may be helpful to run MEN on this dataset and reference the results from their paper. Since the author claims in A1 that the model is not sensitive to the training data, no retraining should be necessary and only inference should be required.
> >
>
> R6:  We understand your question more clearly now. First, we'd like to clarify that Open-Vocabulary Segmentation is not entirely equivalent to the Weakly Supervised 3D-RES setting proposed in our paper.
>
> 1. First, the difference in text length results in varying dependencies on semantic understanding. 3D-RES deals with long texts, requiring more advanced semantic comprehension, which we have specifically addressed in our design. In contrast, Open-Vocabulary Segmentation is limited to short phrases, making it less effective when handling long, open-ended textual descriptions as seen in 3D-RES tasks.
> 2. Furthermore, the focus of our work is to alleviate the burden of mask-level annotations. In our method design, we have drawn inspiration from [17], incorporating category labels. As a result, evaluating the datasets mentioned in Open-Vocabulary Methods using our proposed MEN framework would provide a significant advantage and may not be entirely fair to the Open-Vocabulary Methods. However, if you still require this comparison, please let us know, and we will do our best to provide it before the conclusion of this phase.
>
> [17] Distilling Coarse-to-Fine Semantic Matching Knowledge for Weakly Supervised 3D Visual Grounding. In ICCV 2023.

---

> ### Comment · Reviewer_nJzX · 2024-12-03
>
> We thank the authors for their detailed discussion of these issues. However, many critical concerns remain unresolved.
>
> 1. The method proposed fails fundamentally at handling multiple objects in a scene, showing a 17.5% (@0.25) and 17% (@0.5) drop in accuracy compared to supervised methods when dealing with multiple objects. The authors admit that 60% of their training signals are noisy in these cases. This is not a minor issue - it demonstrates the method cannot handle basic real-world scenarios where multiple similar objects exist in the same scene.
> 2. The method's performance collapses when using different training data, dropping from 33.4 to 24.5 mIoU when switching from ScanNet to S3DIS backbone training. This directly contradicts the paper's main goal of reducing annotation needs across different environments. The authors try to dismiss this by pointing to relative improvements over a baseline, but the absolute performance becomes too poor for practical use.
> 3. The model cannot properly handle spatial relationships between objects. It processes frames one at a time and fails when objects referenced in the text appear in different frames. The authors' claim that this only affects 5% of cases is based on an extremely small number of examples (100) from their dataset. In real-world settings, where objects naturally appear across different viewpoints and spatial relationships are crucial, this architectural limitation would severely impact performance.
>
> Therefore, I believe the work in its current form still needs significant polish to meet the rigorous standards expected at ICLR. The limitations in handling multiple objects, cross-domain generalization, and spatial reasoning need to be more thoroughly addressed.

---

> ### Author Response · Authors · 2024-12-04
> **Response to Reviewer nJzX (part-1)**
>
> > 1. The method proposed fails fundamentally at handling multiple objects in a scene, showing a 17.5% (@0.25) and 17% (@0.5) drop in accuracy compared to supervised methods when dealing with multiple objects. The authors admit that 60% of their training signals are noisy in these cases. This is not a minor issue - it demonstrates the method cannot handle basic real-world scenarios where multiple similar objects exist in the same scene.
> >
>
> A1: Thank you for your valuable feedback. The performance limitations of our approach are most likely due to the two-stage setup, where any inaccuracies or omissions in the first phase can significantly impact the accuracy of the subsequent matching phase, regardless of its intrinsic efficacy, as highlighted in 3D-STMN[1]. Unlike fully-supervised single-stage models, two-stage models are inherently constrained by the segmentation results from the first phase, which often leads to lower overall accuracy.
>
> Nevertheless, when compared to other fully-supervised two-stage methods, such as TGNN[2], InstanceRefer[3], and X-RefSeg3D[4], our approach demonstrates substantial improvements. In particular, our method outperforms InstanceRefer by 4.4 points in Multiple Acc\@0.5, as shown in Table A. This clearly underscores the effectiveness of our model — even without mask supervision, it achieves superior performance compared to fully-supervised methods within the same paradigm, while also effectively handling the “Multiple” cases.
>
> Table A: The 3D-RES results on ScanRefer, where * refers to the results reported by [5].
>
> |  | mIoU | Unique Acc\@0.25 | Unique Acc\@0.5 | Multiple Acc\@0.25 | Multiple Acc\@0.5 | Overall Acc\@0.25 | Overall Acc\@0.5 |
> | --- | --- | --- | --- | --- | --- | --- | --- |
> | TGNN | 27.8 | - | - | - | - | 37.5 | 31.4 |
> | TGNN* | 28.8 | 69.3 | 57.8 | 31.2 | 26.6 | 38.6 | 32.7 |
> | InstanceRefer* | 30.6 | 81.6 | 72.2 | 29.4 | 23.5 | 40.2 | 33.5 |
> | X-RefSeg3D  | 29.9 | - | - | - | - | 40.3 | 33.8 |
> | ours | 33.4 | 83.1 | 79.1 | 32.6 | 27.9 | 42.4 | 37.8 |
>
> [1] 3D-STMN: Dependency-Driven Superpoint-Text Matching Network for End-to-End 3D Referring Expression Segmentation. In AAAI 2024.
>
> [2] Text-Guided Graph Neural Networks for Referring 3D Instance Segmentation. In AAAI 2021.
>
> [3] InstanceRefer: Cooperative Holistic Understanding for Visual Grounding on Point Clouds through Instance Multi-level Contextual Referring. In ICCV 2021.
>
> [4] X-RefSeg3D: Enhancing Referring 3D Instance Segmentation via Structured Cross-Modal Graph Neural Networks. In AAAI 2024.
>
> [5] 3D-GRES: Generalized 3d referring expression segmentation. In ACM MM 2024.
>
> > 2. The method's performance collapses when using different training data, dropping from 33.4 to 24.5 mIoU when switching from ScanNet to S3DIS backbone training. This directly contradicts the paper's main goal of reducing annotation needs across different environments. The authors try to dismiss this by pointing to relative improvements over a baseline, but the absolute performance becomes too poor for practical use.
> >
>
> A2: Thanks for your feedback. First of all, we fully understand your concerns regarding the performance discrepancy of the model across different datasets. Regarding the issue you mentioned, where the mIoU performance drops from 33.4 on ScanNet to 24.5 when training the backbone network from ScanNet to S3DIS, this decline in performance is actually consistent with objective reality. As the well-known saying in machine learning goes, "There is no free lunch." On the S3DIS dataset, there are only 12 classes, whereas ScanNet provides 20 classes. Transferring a backbone trained on this dataset to S3DIS inevitably leads to a performance drop.
>
> In theory, no method can achieve better or equivalent performance in this setting, as it contradicts the *No Free Lunch Theorem*. We believe that transferability should be demonstrated when moving from a richer dataset to a data-scarce domain, where good performance can still be achieved. However, it is unreasonable to expect the same performance when transferring to a domain with lower-quality data. We hope our explanation clarifies this issue to your satisfaction.

---

> ### Author Response · Authors · 2024-12-04
> **Response to Reviewer nJzX (part-2)**
>
> > 3. The model cannot properly handle spatial relationships between objects. It processes frames one at a time and fails when objects referenced in the text appear in different frames. The authors' claim that this only affects 5% of cases is based on an extremely small number of examples (100) from their dataset. In real-world settings, where objects naturally appear across different viewpoints and spatial relationships are crucial, this architectural limitation would severely impact performance.
> >
>
> A3: Thanks for your insightful feedback. In fact, our model is capable of properly handling spatial relationships between objects. As shown in Table 2, our approach outperforms the method that only considers the category and attributes of the target object (row 6 in Table 2) by 7.5 points in overall performance, and by 9.1 points in the *Multiple* scenario.
>
> This performance improvement stems from our model's full-context expert, which is better equipped to leverage the visual context of the target, including its relationships with nearby objects. Additionally, it is also more adept at understanding the complex relational descriptions within the referring expression.
>
> This provides strong evidence of our method’s capability to comprehend the spatial relationships between objects mentioned in the description. We appreciate your valuable feedback.

---

### Official Review · Reviewer_kMXL · 2024-10-30

**Soundness:** 2
**Presentation:** 2
**Contribution:** 3
**Rating:** 5
**Confidence:** 4

**Summary:**

The paper proposes a weakly supervised 3D-RES task and introduces the Multi-Expert Network (MEN), a framework that leverages multiple cues for referring expression reasoning, reducing reliance on expensive mask-description pair annotations. The framework includes the Multi-Expert Mining (MEM) module, which aligns visual and textual contexts from three complementary cues, and the Multi-Expert Aggregation (MEA) module, which consolidates expert knowledge while filtering out interference to emphasize the target instance. Extensive experiments on the ScanRefer and Multi3DRefer datasets validate the effectiveness of our approach in addressing weakly-supervised 3D-RES.

**Strengths:**

1. Proposes a novel weakly supervised approach to handle 3D RES.
2. The paper is well-written and easy to understand.
3. Demonstrates impressive results across multiple metrics in the weakly supervised setting.

**Weaknesses:**

1. This pipeline processes each frame separately, however, target objects and anchors are not always present within the same frame due to the limited field of view. This slightly goes against the authors' statement in Lines 189-191: “The full-context expert focuses on... understanding the spatial relationships and the surrounding environment of the target.”
2. The proposed pipeline appears to struggle with cases containing zero targets, even in scenarios without distractors, as shown in Table 2.
3. The ablation study in Table 3 is limited to evaluating the performance of individual modules. Specifically, it would be helpful to see the performance of samples in the "Unique" category.

**Questions:**

1. How can one ensure that the methods effectively address the issues outlined in Weakness #1?
2. Have you considered adding a confidence threshold to allow the pipeline to handle scenarios with zero targets?
3. Could you provide the ablation studies mentioned in Weakness #3?

---

> ### Author Response · Authors · 2024-11-24
> **Official Comment by Authors (part-1)**
>
> > **W1: This pipeline processes each frame separately, however, target objects and anchors are not always present within the same frame due to the limited field of view. This slightly goes against the authors' statement in Lines 189-191: “The full-context expert focuses on... understanding the spatial relationships and the surrounding environment of the target.”**
> >
> > **Q1: How can one ensure that the methods effectively address the issues outlined in Weakness #1?**
>
>
> A1: Thank you for your constructive question. While the scenario you mentioned is meaningful and interesting, it is relatively rare in the ScanRefer dataset due to annotation limitations. To verify this, we conducted an analysis by randomly sampling 100 examples from the ScanRefer validation set. Through manual inspection, we checked whether the 2D frames selected by the model included all the targets and anchors referenced in the text descriptions. Results showed that 95% of the samples contained all the relevant objects, indicating that the selected frames generally capture the full visual context described in the text. We have added this discussion to Section C in the updated version.
>
> This finding aligns with the experimental results in Table 3 (rows 2 and 4), where incorporating the *full-context expert* significantly improved performance. Specifically, the *full-context expert*, which captures spatial relationships and contextual information, increased the mIoU by 15.4 points compared to using only the *attribute expert*, which focuses solely on target-specific features.
>
> We recognize the significance of the issue you raised and its potential implications. As part of our future work, we plan to enhance the model's robustness to handle scenarios where some anchors may be missing from the selected frames. Additionally, we aim to expand the dataset to include more such challenging examples, providing a new benchmark for further research. Thank you again for bringing attention to this critical aspect.
>
> > **W2: The proposed pipeline appears to struggle with cases containing zero targets, even in scenarios without distractors, as shown in Table 2.**
> >
> > **Q2: Have you considered adding a confidence threshold to allow the pipeline to handle scenarios with zero targets?**
>
> A2:  Thank you for your insightful question. As mentioned in Lines 362–365, our current pipeline combines the masks of proposals with probabilities greater than 0.1 to form the predicted mask. If no proposals exceed this threshold, the case is predicted as zero-target. To evaluate this mechanism, we analyzed the impact of varying thresholds on 3D-GRES performance, as presented in Table 10 of the supplementary materials. While increasing the threshold improved zero-target (zt) detection, it negatively affected performance on multi-target (mt) cases, showing that threshold adjustments alone cannot effectively address the zero-target problem.
>
> Inspired by your feedback, we explored an additional approach to improve zero-target detection without compromising performance in other cases. Specifically, we introduced a dedicated *zt_head*, treating zero-target detection as a classification task compatible with our weakly supervised setup. It uses only category labels without adding expensive mask labels, as explained in section D of the updated paper.
>
> The results demonstrate that the *zt_head* significantly improved performance on zero-target cases, without negatively impacting other scenarios. This enhancement highlights the robustness of the proposed pipeline.
>
> We sincerely appreciate your suggestion and have included this adjustment in the updated version (highlighted in blue, lines 365–367, 429-431, 945-977). This improvement not only enriches the paper but also addresses an important challenge in the 3D-GRES task. Thank you again for your thoughtful input.
>
> Table a: Results of 3D-GRES, including mIoU and Acc\@0.5, where "w/o zt_head" means that the zt_head was not used, and "w/ zt_head" means that the zt_head was used to predict whether it is a zero target.
>
> |  | mIoU | zt  w/  dis | zt  w/o  dis | st  w/ dis | st  w/o dis | mt | Overall |
> | --- | --- | --- | --- | --- | --- | --- | --- |
> | w/o zt_head | 30.6 | 1.3 | 0.4 | 28.9 | 59.5 | 19.8 | 30.1 |
> | w/ zt_head | 33.4 | 22.2 | 55.9 | 28.7 | 59.4 | 19.7 | 33.4 |

---

> > ### Comment · Reviewer_kMXL · 2024-11-25
> >
> > 1. In your response A1, you demonstrated that such cases do not always occur; however, they do exist. A suggestion, to address this problem, incorporating the "appearance of the anchors" as criteria in your "Visibility-based Selection" is a straightforward solution.
> >
> > 2. In your response A2, the introduction of the Zero Target Head in the experiments seems abrupt. It would be beneficial to introduce it earlier. If space is a concern, including this result in the appendix would also be helpful if you think it's not part of your main proposed solution.
> >
> > 3. Thank you for providing Table B with the Unique category column. However, I noticed that the results in Table B contradict those in Table 3 of your updated manuscript. Could you please clarify their difference and which table I should refer to so I can provide accurate feedback?

---

> ### Author Response · Authors · 2024-11-24
> **Official Comment by Authors (part-2)**
>
> > **W3: The ablation study in Table 3 is limited to evaluating the performance of individual modules. Specifically, it would be helpful to see the performance of samples in the "Unique" category.**
> >
> > **Q3: Could you provide the ablation studies mentioned in Weakness #3**
> >
>
> A3: Thank you for your valuable suggestion to expand our ablation studies. In the updated version of the paper (see Tables b and c), we have included the performance of samples in the "Unique" category.
>
> The results reveal that the performance challenges in the "Unique" category arise from ambiguity in matching when multiple instances are present within the selected 2D frames, which can hinder the *full-context expert*. However, the *cls expert*, which leverages category-specific information, demonstrates greater effectiveness in resolving these ambiguities. This finding underscores the complementary strengths of the individual experts and highlights the importance of our multi-expert approach.
>
> We sincerely appreciate your constructive feedback, as it has enhanced the clarity and comprehensiveness of our analysis while validating the necessity of the proposed modules. Thank you again for helping us improve the quality of the paper.
>
> Table b: Ablation studies on MEM, where “full-cont.” means full-context expert, “attri.” means attribute expert, “cate.” means category expert.
>
> |  | full-cont. | attri. | cate. | mIoU | Unique Acc\@0.5 | Multiple Acc\@0.5 | Overall Acc\@0.5 |
> | --- | --- | --- | --- | --- | --- | --- | --- |
> |  |  |  |  |  | 0.5 | 0.5 | 0.5 |
> | 1 | $\checkmark$ |  |  | 23.4 | 51.2 | 19.8 | 25.9 |
> | 2 |  | $\checkmark$ |  | 10.8 | 23.2 | 9.5 | 12.2 |
> | 3 |  |  | $\checkmark$ | 25.6 | 78.8 | 17.4 | 29.3 |
> | 4 | $\checkmark$ | $\checkmark$ |  | 26.2 | 58.6 | 22.4 | 29.4 |
> | 5 | $\checkmark$ |  | $\checkmark$ | 32.9 | 78.3 | 27.1 | 37 |
> | 6 |  | $\checkmark$ | $\checkmark$ | 26.4 | 78 | 18.8 | 30.3 |
> | 7 | $\checkmark$ | $\checkmark$ | $\checkmark$ | 33.4 | 79.1 | 27.9 | 37.8 |

---

> ### Author Response · Authors · 2024-11-25
> **Response to Reviewer kMXL**
>
> > A1: In your response A1, you demonstrated that such cases do not always occur; however, they do exist. A suggestion, to address this problem, incorporating the "appearance of the anchors" as criteria in your "Visibility-based Selection" is a straightforward solution.
> >
>
> R1: Thank you very much for your suggestion. The approach you proposed is highly beneficial for refining the frames. We will update this experiment as quickly as possible within the limited time available. Once again, thank you for your patience and thoughtful input.
>
> > A2: In your response A2, the introduction of the Zero Target Head in the experiments seems abrupt. It would be beneficial to introduce it earlier. If space is a concern, including this result in the appendix would also be helpful if you think it's not part of your main proposed solution.
> >
>
> R2: Thank you very much for your kind suggestion. We will include a detailed description of this section in the supplementary materials. The updated version will be finalized before the end of this phase.
>
> > A3: Thank you for providing Table B with the Unique category column. However, I noticed that the results in Table B contradict those in Table 3 of your updated manuscript. Could you please clarify their difference and which table I should refer to so I can provide accurate feedback?
> >
>
> R3: We apologize for the confusion. The results in Table 3 of the updated manuscript were mistakenly filled in due to haste. Please refer to Table B for the correct results. Additionally, we have revised the "Unique" results in Table 3 of the updated version to align them with those in Table B. We sincerely appreciate your careful attention to detail, which has helped us ensure the accuracy and clarity of our work.

---

> > ### Comment · Reviewer_kMXL · 2024-11-28
> >
> > 1. I find the implementation details of the new zt_head module a bit unclear. For example, in Table 2 of the paper, while the overall performance improves with the addition of zt_head, its impact on the performance of other columns is not fully explained. It would be helpful to clarify how the results from the zt_head are integrated and utilized with the MEM modules. This aspect deserves more detailed discussion in the manuscript to provide a clearer understanding of the interactions between components.
> >
> > 2. The current version of the manuscript is somewhat unclear regarding the experimental setup with the zt_head. Since this module has been incorporated into the main methodology, it is important to provide more details in your experiments. For instance, it is not entirely clear whether the zt_head was included in the ablation studies. You might have to update the explanations to include a discussion of zt_head for the experimental settings.
> >
> > 3. As the manuscript deadline approaches, the current version still requires significant refinement before it can be considered for publication. I would strongly suggest reorganizing and clarifying the sections related to the added details.  A more polished and comprehensive revision would be essential before the submission deadline.

---

> ### Author Response · Authors · 2024-11-28
> **Response to Reviewer kMXL**
>
> Thank you very much for your feedback. Before receiving your latest response, we have already updated the paper based on your suggestions. We are just refining it a bit more before submitting the final PDF, so it hasn't been uploaded to the system yet. Before addressing your latest feedback, we wanted to share the current version with you in hopes of receiving your valuable input before the submission deadline.
>
> In this version, we’ve consolidated the details and experimental structure of the 3D GRES into one section and placed everything in the supplementary materials under Section D. The ZERO-TARGET DETECTION HEAD has been specifically placed in subsection D.4.
>
> Moving forward, we will carefully make the necessary adjustments and respond to your latest feedback.

---

> ### Author Response · Authors · 2024-11-28
> **Response to Reviewer kMXL**
>
> > 1. I find the implementation details of the new zt_head module a bit unclear. For example, in Table 2 of the paper, while the overall performance improves with the addition of zt_head, its impact on the performance of other columns is not fully explained. It would be helpful to clarify how the results from the zt_head are integrated and utilized with the MEM modules. This aspect deserves more detailed discussion in the manuscript to provide a clearer understanding of the interactions between components.
> >
>
> A1: Thank you for your valuable suggestion. In response, we have provided a detailed description of the implementation of the zt_head module and its integration with the proposed MEM and MEA modules in Section D.1, D.2, lines 963-999, and Figure 5. Specifically, we decouple the zero-target task using this branch, which runs in parallel with MEM to predict whether the current sample is a zero-target. If it is, the result is directly set as zero-target; if not, we filter the MEM and MEA proposals using a threshold and concatenate the masks above the threshold as the final prediction.
>
> Regarding the " impact on the performance of other columns," thanks to this decoupling, the accuracy for zero-target cases improves significantly. At the same time, the accuracy for other columns, such as single and multi-target scenarios, does not decrease. This is because the zt_head rarely misclassifies a target as zero-target, so it does not reduce the recall for single and multi-target cases, thus maintaining robustness for those scenarios. We have included this discussion in Section D.4, lines 1017-1020, and 1030-1035 of the manuscript. Thank you again for your insightful feedback.
>
> > 2. The current version of the manuscript is somewhat unclear regarding the experimental setup with the zt_head. Since this module has been incorporated into the main methodology, it is important to provide more details in your experiments. For instance, it is not entirely clear whether the zt_head was included in the ablation studies. You might have to update the explanations to include a discussion of zt_head for the experimental settings.
> >
>
> A2: Thank you for your thoughtful guidance and for pointing out this issue. In response to your suggestions, we have moved the content related to the 3D-GRES task (including the zt_head) to the supplementary materials for clearer presentation. As a result, the main body of the manuscript, prior to the references, now focuses exclusively on experiments related to the traditional 3D-RES task, excluding any zero-target scenarios, as mentioned in lines 957-961 of the updated version. Additionally, in Section D of the supplementary materials, we have provided a detailed description of the experimental setups, particularly with regard to the zt_head discussion. Specifically, we have added an ablation study on the zt_head, which can be found in Section D.4, lines 1015-1020, 1030-1035, and Table 13, 14. These changes aim to enhance the rigor and clarity of our paper. Once again, thank you for your valuable feedback and suggestions.
>
> > 3. As the manuscript deadline approaches, the current version still requires significant refinement before it can be considered for publication. I would strongly suggest reorganizing and clarifying the sections related to the added details. A more polished and comprehensive revision would be essential before the submission deadline.
> >
>
> A3: Thank you for your valuable feedback. In response to your suggestions, we have made improvements to the manuscript, particularly in integrating the new details, such as those related to the zt_head, more cohesively with the rest of the paper. We will continue refining the manuscript to ensure it is polished and clear before the final version. As you mentioned, your feedback has been instrumental in helping us make the paper more readable and comprehensible, and we hope our revisions meet your expectations.

---

> > ### Comment · Reviewer_kMXL · 2024-12-03
> >
> > In Appendix E, you mention projecting the 3D masks onto 2D frames to evaluate visibility. Additionally, in Appendix C (lines 905–907), you state that the ScanRefer dataset lacks explicit annotations for all anchors and their presence in specific frames. From this, I infer that you are using the annotated target 3D mask and projecting it onto 2D frames to determine visibility. Is this correct?

---

> ### Author Response · Authors · 2024-12-03
> **Response to Reviewer kMXL**
>
> > In Appendix E, you mention projecting the 3D masks onto 2D frames to evaluate visibility. Additionally, in Appendix C (lines 905–907), you state that the ScanRefer dataset lacks explicit annotations for all anchors and their presence in specific frames. From this, I infer that you are using the annotated target 3D mask and projecting it onto 2D frames to determine visibility. Is this correct?
> >
>
> A: Thank you for your attention to detail. We do not use the annotated target 3D mask to determine visibility. Instead, as stated in Lines 213–215, the 3D masks we project are proposal masks generated by SoftGroup segmentation. We select the 2D frames with the top 15 highest visibility for each proposal as its visual context. Neither during training nor inference do we utilize the annotated target 3D mask.
>
> This approach does not contradict our statement in Appendix C regarding the lack of annotations for anchors. We will rephrase this explanation in the updated version of our paper to improve clarity and hope this response resolves your concerns.

---

> ### Comment · Reviewer_kMXL · 2024-12-03
>
> 1. Methodology for Verifying Anchor Objects: The approach of randomly sampling 100 instances from ScanRefer and conducting manual inspections may not provide a sufficiently rigorous or comprehensive analysis. More systematic and reliable methods could be employed to improve the validity of the verification process:
>
>         a) Consider utilizing a state-of-the-art 2D detection model to automatically identify anchor objects within the scenes. This would provide a more scalable and efficient verification approach.
>
>         b) Apply the same techniques used to obtain the 3D masks for target objects to also extract and verify the anchors. This would help ensure consistency across object types.
>
>         c) Leverage a Large Language Model (LLM) to extract anchor object classes from the textual descriptions, and then cross-reference these with ground truth 3D bounding box annotations to verify the presence of the anchor objects in the frames. This method could offer a more robust and automated verification process.
>
> 2. Even if the conclusion that only 5% of the samples lack anchor objects accurately reflects the data distribution in ScanRefer, the underlying concern raised in W1 and Q1 remains valid. This suggests that your method may overlook anchor objects mentioned in the text, focusing instead on detecting target objects in isolation. Such an approach might lead to the arbitrary selection of a target object when multiple objects of the same class are present in a scene. Evidence of this issue can be seen in Table 1 and Table 3, where your method seems to struggle with samples that contain multiple instances of the target class (denoted as "Multiple"). For example in Table 1, compared to supervised methods, your best-performing method only falls short by 4.6% in IoU≥0.5 "Unique" samples, however, the "Multiple" samples have a significant drop of 17%.
>
> While the authors have provided some clarifications in the manuscript and in response to reviewer comments, these explanations seem to underscore the issues I initially raised. In fact, these clarifications seem to highlight the issues and weaknesses I raised, which further supports the validity of my concerns.

---

> ### Author Response · Authors · 2024-12-04
> **Response to Reviewer kMXL (part-1)**
>
> > 1. Methodology for Verifying Anchor Objects: The approach of randomly sampling 100 instances from ScanRefer and conducting manual inspections may not provide a sufficiently rigorous or comprehensive analysis. More systematic and reliable methods could be employed to improve the validity of the verification process:
> >
> >
> > ```
> >  a) Consider utilizing a state-of-the-art 2D detection model to automatically identify anchor objects within the scenes. This would provide a more scalable and efficient verification approach.
> >
> >  b) Apply the same techniques used to obtain the 3D masks for target objects to also extract and verify the anchors. This would help ensure consistency across object types.
> >
> >  c) Leverage a Large Language Model (LLM) to extract anchor object classes from the textual descriptions, and then cross-reference these with ground truth 3D bounding box annotations to verify the presence of the anchor objects in the frames. This method could offer a more robust and automated verification process.
> > ```
> >
> R1: Thank you for your feedback. Based on your suggestions, we have conducted relevant statistics on the entire ScanRefer validation set, which will be included in the final version of the paper. Specifically, inspired by your feedback, we sought to systematically validate the coverage of anchor objects in the selected 2D frames through the following steps:
>
> 1. **Using an LLM (GPT-3.5-turbo) to assess and rewrite natural language descriptions.**
>
>     Specifically, we first instructed the LLM to determine whether the natural language description contained information about the anchor. If not, the description was skipped. If it did, the description was rewritten to set the previous anchor as the target and the previous target as the anchor. The detailed prompt is provided in Table A.
>
> 2. **Using a multimodal large model (GPT-4o) to verify the presence of the anchor in the selected 2D frames.**
>
>     In this step, we input the selected 2D frames and the rewritten anchor-specific description into the multimodal model, which then determined whether the 2D frame contained the anchor. The detailed prompt is provided in Table B.
>
> 3. **Collecting statistics on all samples, including whether the text descriptions contained an anchor and whether the selected 2D frames contained the anchor.**
>
> Following this process, we found that in 96.1% of cases, the selected 2D frames already included all objects described in the text, including both the target and the anchor (if present). We hypothesize that this is because humans tend to describe targets using nearby objects as anchors.
>
> Moreover, inspired by your suggestions, we plan to further enhance our model's ability to perceive challenging cross-frame anchors in the journal version of this paper. Our planned extensions include the following:
>
> 1. Employing state-of-the-art 2D RES models to locate potential targets and anchors in 2D frames. Specifically, we will use the target description and the rewritten anchor description as inputs to obtain candidate masks for targets and anchors in the 2D frames.
> 2. Projecting the 2D candidate targets and anchors back into the 3D point cloud scene. We will pair these projected candidates to render images where both the target and anchor appear, generating N target-anchor paired renderings.
> 3. Filtering rendered images based on the visibility of each 3D instance proposal in the target-anchor paired renderings using the method outlined in Appendix E. We will then compute the similarity between the selected rendered images and the target description text.
> 4. Incorporating this expert-generated probability distribution into the MEN model via MEA fusion. This extension aims to improve MEN’s cross-frame anchor perception capabilities.
>
> We sincerely thank you for your valuable suggestions, which have significantly improved our paper and will further advance progress in this field.

---

> ### Author Response · Authors · 2024-12-04
> **Response to Reviewer kMXL (part-2)**
>
> **Supplement to A1:**
>
> Table A: Prompt Used for Assessing and Rewriting Descriptions.
>
> ```json
> {
> 		"role":"system",
> 		"content":
> 		    "You are an intelligent chatbot designed to extract attribute information of a target object from a given sentence."
> 		    "Your task is to analyze the sentence I provide, which describes a target object."
> 		    "You need to determine whether there is an anchor object in the sentence that helps in locating the target object."
> 		    "If an anchor object exists, you should treat all anchor objects as a whole and modify the sentence to make the anchor objects the target and the target object the anchor."
> 		    "Note that when there is a spatial relationship between the target and the anchor, you should modify the descriptive terms indicating their positions."
> 		    "For example, if the target is to the left of the anchor, then the anchor should be described as being to the right of the target; if the target is above the anchor, then the anchor should be described as being below the target."
> },
> {
> 	  "role":"user",
> 	  "content":
> 	      "Please determine whether there is an anchor object in the input sentence."
> 	      "If an anchor object exists, rewrite the sentence as required."
> 	      "If no anchor object exists, no changes should be made to the sentence."
> 	      f"Input sentence: {text}\n"
> 	      "DO NOT PROVIDE ANY OTHER OUTPUT TEXT OR EXPLANATION. Only provide the Python dictionary string."
> 	      "------"
> 	      "Here are some examples:"
> 	      "Input sentence: There is a silver bread toaster. Placed next to the fridge."
> 	      "Your response should look like this: {'anchor_exist': 1, 'anchor_description': 'The fridge is next to the silver bread toaster.'}."
> 	      "Input sentence: There is a beige wooden working table."
> 	      "Your response should look like this: {'anchor_exist': 0, 'anchor_description': ''}."
> 	      "Input sentence: A coffee table with light purple color surrounded by a light purple sofa."
> 	      "Your response should look like this: {'anchor_exist': 1, 'anchor_description': 'The light purple sofa surrounds the coffee table'}."
> 	      "Input sentence: There is a white toilet."
> 	      "Your response should look like this: {'anchor_exist': 0, 'anchor_description': ''}."
> }
> ```
>
> Table B: Prompt Used for Determining the Presence of an Anchor in the Selected 2D Frames.
>
> ```json
> {
>     "role": "user",
>     "content": [
>     {"type": "text", "text": "I will provide you with a sentence. Please determine whether the image contains the target object described in the sentence."
>      f"Input sentence: {text}\n."
>      "DO NOT PROVIDE ANY OTHER OUTPUT TEXT OR EXPLANATION. Only provide the Python dictionary string."
>      "For example, if the image contains the target, your response should look like this: {'score': 1};"
>      "if the image does not contain the target, your response should look like this: {'score': 0}."
>     },
>     {
>         "type": "image_url",
>         "image_url": {
>         "url":  "URL-of-the-image"
>         },
>     },
>     ],
> }
> ```

---

> ### Author Response · Authors · 2024-12-04
> **Response to Reviewer kMXL (part-3)**
>
> > 2. Even if the conclusion that only 5% of the samples lack anchor objects accurately reflects the data distribution in ScanRefer, the underlying concern raised in W1 and Q1 remains valid. This suggests that your method may overlook anchor objects mentioned in the text, focusing instead on detecting target objects in isolation. Such an approach might lead to the arbitrary selection of a target object when multiple objects of the same class are present in a scene. Evidence of this issue can be seen in Table 1 and Table 3, where your method seems to struggle with samples that contain multiple instances of the target class (denoted as "Multiple"). For example in Table 1, compared to supervised methods, your best-performing method only falls short by 4.6% in IoU≥0.5 Unique samples, however, the hard samples have a significant drop of 17%.
> >
>
> A2: Thank you for your feedback. In fact, the current approach has already considered different anchors; however, further research may be required for anchors that are more distant. Specifically, to address the understanding of different anchors in long expressions, we introduced LongCLIP to comprehend fine-grained textual information. This approach has proven to be effective, as demonstrated below:
>
> (1) **Our method does not rely on the arbitrary selection of a target object when multiple objects of the same class are present in a scene.**
>
> This is reflected in Table 2. The third row corresponds to the approach of selecting a target purely based on class (i.e., arbitrary selection of a target object when multiple objects of the same class are present in a scene).
>
> Clearly, the performance of our model (row 7) significantly surpasses this random selection approach based solely on class, especially in ”Multiple“ scenarios, where the improvement is 10.5 points. This highlights the effectiveness of our model in handling ”Multiple“ cases by leveraging the visual context of the target and the complex spatial relationships described in the referring expression to distinguish the target from other distractors of the same class.
>
> (2) **Compared to other two-stage fully-supervised methods such as TGNN[1], InstanceRefer[2], and X-RefSeg3D[3], our method achieves superior performance across the board.**
>
> Notably, in terms of **Multiple Acc\@0.5**, our method surpasses InstanceRefer by 4.4 points. This strongly demonstrates the superiority of our model, which achieves performance exceeding that of fully-supervised methods of the same paradigm, even without mask supervision, while effectively handling ”Multiple“ cases.
>
> As the first work to propose this research direction, our method can serve as a strong baseline for future studies. It is both straightforward to follow and makes substantial contributions to the community.
>
> Table C: The 3D-RES results on ScanRefer, where * refers to the results reported by [4].
>
> |  | mIoU | Unique Acc\@0.25 | Unique Acc\@0.5 | Multiple Acc\@0.25 | Multiple Acc\@0.5 | Overall Acc\@0.25 | Overall Acc\@0.5 |
> | --- | --- | --- | --- | --- | --- | --- | --- |
> | TGNN | 27.8 | - | - | - | - | 37.5 | 31.4 |
> | TGNN* | 28.8 | 69.3 | 57.8 | 31.2 | 26.6 | 38.6 | 32.7 |
> | InstanceRefer* | 30.6 | 81.6 | 72.2 | 29.4 | 23.5 | 40.2 | 33.5 |
> | X-RefSeg3D  | 29.9 | - | - | - | - | 40.3 | 33.8 |
> | ours | 33.4 | 83.1 | 79.1 | 32.6 | 27.9 | 42.4 | 37.8 |
>
> [1] Text-Guided Graph Neural Networks for Referring 3D Instance Segmentation. In AAAI 2021.
>
> [2] InstanceRefer: Cooperative Holistic Understanding for Visual Grounding on Point Clouds through Instance Multi-level Contextual Referring. In ICCV 2021.
>
> [3] X-RefSeg3D: Enhancing Referring 3D Instance Segmentation via Structured Cross-Modal Graph Neural Networks. In AAAI 2024.
>
> [4] 3D-GRES: Generalized 3d referring expression segmentation. In ACM MM 2024.

---

### Official Review · Reviewer_tjsR · 2024-10-31

**Soundness:** 3
**Presentation:** 3
**Contribution:** 3
**Rating:** 6
**Confidence:** 4

**Summary:**

This paper introduces the Multi-Expert Network (MEN), a weakly supervised framework for 3D Referring Expression Segmentation (3D-RES) that reduces dependency on costly annotations. MEN uses two modules: Multi-Expert Mining (MEM) to extract semantic cues from various dimensions and Multi-Expert Aggregation (MEA) to consolidate these cues.

**Strengths:**

Experimental results demonstrate MEN’s effectiveness in accurately segmenting 3D objects based on text descriptions, even with limited supervision, outperforming some fully supervised 3D-RES methods.

**Weaknesses:**

1. The paper lacks a discussion of related works, specifically on open-vocabulary 3D segmentation. Open-vocabulary 3D segmentation techniques utilize open-vocabulary queries (such as class labels or text queries) to segment 3D objects. These approaches can be adapted for weakly supervised 3D referring segmentation by replacing open-vocabulary queries with referring expressions. For example, methods [A-C] could be applied to weakly-supervised 3D segmentation tasks similar to those proposed in this paper, as their models operate without manual 3D annotations or can produce 3D segmentation results in a zero-shot manner. Including these methods in the related work discussion or comparing them in the experiments would strengthen the paper.
2. Methods [A-C] are capable of handling weakly-supervised 3D referring segmentation.
3. The writing is hard to follow, particularly in Lines 294-323. For instance, the phrase in Line 299, “Then &(P1, …, Pn) exists,” lacks clarity, making its intent unclear. Furthermore, Eq. (13) includes a parameter $\sigma_x$, which is missing in Eq. (14).
4. The symbols in Eq. (11) are inconsistent with those in Figure 2. Eq (11) uses $\cdot$ to denote matrix multiplication, but Figure 2 uses $\otimes$.

Minor issue:
Incorrect description in Figure 4(a): It appears that the target in Figure 4(a) is a table, not a couch, yet the descriptions for (a) and (b) are identical.

[A] OpenScene: 3D Scene Understanding with Open Vocabularies. In CVPR 2023

[B] OpenMask3D: Open-Vocabulary 3D Instance Segmentation. In NIPS 2023.

[C] Open3DIS: Open-Vocabulary 3D Instance Segmentation with 2D Mask Guidance. In CVPR 2024

**Questions:**

In the category expert, if the class label can be directly extracted from the input text, why is a classifier needed for the text as well?

---

> ### Author Response · Authors · 2024-11-24
> **Official Comment by Authors (part-1)**
>
> > **Q1: The paper lacks a discussion of related works, specifically on open-vocabulary 3D segmentation. Open-vocabulary 3D segmentation techniques utilize open-vocabulary queries (such as class labels or text queries) to segment 3D objects. These approaches can be adapted for weakly supervised 3D referring segmentation by replacing open-vocabulary queries with referring expressions. For example, methods [A-C] could be applied to weakly-supervised 3D segmentation tasks similar to those proposed in this paper, as their models operate without manual 3D annotations or can produce 3D segmentation results in a zero-shot manner. Including these methods in the related work discussion or comparing them in the experiments would strengthen the paper.**
> >
> >
> > [A] OpenScene: 3D Scene Understanding with Open Vocabularies. In CVPR 2023
> >
> > [B] OpenMask3D: Open-Vocabulary 3D Instance Segmentation. In NIPS 2023.
> >
> > [C] Open3DIS: Open-Vocabulary 3D Instance Segmentation with 2D Mask Guidance. In CVPR 2024
> >
>
> A1: Thank you for your valuable feedback. We fully agree that incorporating open-vocabulary 3D segmentation models into our discussion and experiments would significantly enhance the paper.
>
> To address your suggestion:
>
> 1. We have added a detailed discussion of related works in the updated version. (highlighted in blue, lines 127–136, 409-414).
> 2. Additionally, we evaluated the performance of weakly-supervised 3D-RES on the ScanRefer dataset using the official code repositories of [A], [B], and [C]. The results are summarized in Table a.
>
> In the next version of the paper, we will integrate these discussions into the main manuscript, providing a more comprehensive analysis. This not only situates our work more robustly within the existing literature but also underscores the superior performance of our method in the weakly-supervised 3D-RES task.
>
> Table a: Performance Comparison with Open Vocabulary Methods on ScanRefer.
>
> |  | mIoU | Unique Acc\@0.25 | Unique Acc\@0.5 | Multiple Acc\@0.25 | Multiple Acc\@0.5 | Overall Acc\@0.25 | Overall Acc\@0.5 |
> | --- | --- | --- | --- | --- | --- | --- | --- |
> | OpenScene | 12.0 | 35.2 | 12.4 | 8.6 | 1.6 | 13.8 | 3.7 |
> | OpenMask3D | 12.8 | 20.8 | 19.8 | 13.9 | 12.4 | 15.2 | 13.8 |
> | Open3DIS | 12.3 | 22.3 | 11.9 | 13.8 | 5.9 | 15.4 | 7.1 |
> | Ours | 33.4 | 83.1 | 79.1 | 32.6 | 27.9 | 42.4 | 37.8 |
>
> > **Q2: Methods [A-C] are capable of handling weakly-supervised 3D referring segmentation.**
> >
>
> A2: Thank you for your feedback. While open-vocabulary methods ([A-C]) can be applied to weakly-supervised 3D-RES tasks, their performance is suboptimal, particularly in complex scenarios like the "multiple" setting. These methods rely on single-cue textual features from encoders like CLIP, focusing on visual features while neglecting the complexity of lengthy and hierarchical textual descriptions or the nuanced multimodal alignment needed for 3D-RES, as highlighted in [D].
>
> Experimental results confirm that their performance is comparable to models using only the attribute expert (Table 3, Row 2), indicating limited ability to process fine-grained textual details. In contrast, our **Multiple Expert Mining module** addresses these shortcomings by integrating multiple alignment cues. Specifically, the **full context expert** leverages LongCLIP for better understanding of complex textual descriptions, while multi-expert fusion enables robust handling of both coarse and fine-grained details.
>
> As a result, our model achieves significant improvements, including a 15.5-point gain over OpenMask3D in Acc\@0.5 for “Multiple”, demonstrating the strength of our approach. Relevant results are included in the Table 1. We appreciate your constructive feedback, which has helped us better present our contributions.
>
> [D] Described object detection: Liberating object detection with flexible expressions. In NeurIPS 2024.
>
> > **Q3: The writing is hard to follow, particularly in Lines 294-323. For instance, the phrase in Line 299, “Then &(P1, …, Pn) exists,” lacks clarity, making its intent unclear.**
> >
>
> A3: Thank you for your attention to detail. As shown in Lines 294-25, “Then &(P1, …, Pn) exists,” refers to the consolidation of a finite number of probability distributions P1, …, Pn. In this section, we refer to Theorem 3.1 from [E] and have ensured consistency with the original description. To improve the clarity of our paper, we have included the proof process for this theorem in Section G of the updated version.
>
> [E] Hill T. Conflations of probability distributions[J]. Transactions of the American Mathematical Society, 2011, 363(6): 3351-3372.

---

> ### Author Response · Authors · 2024-11-24
> **Official Comment by Authors (part-2)**
>
> > **Q4: Furthermore, Eq. (13) includes a parameter** $\delta_x$**, which is missing in Eq. (14).**
> >
>
> A4: Thank you for your careful observation. The parameter $\delta_x$  is not missing in Eq. (14) but is implicitly set to 1. Below is an explanation for this:
>
> As described in Lines 305–306, “$\delta_x$ is the Dirac delta measure[F] in the set of all real Borel probability measures[G] at the point $x$ (i.e., $\delta_x(B)  = 1$ if $x\in B$, and $= 0$ if $x\notin B$).” In Line 312, it is used to ensure that the random variable $x$ always lies within the common atoms $A$. Specifically, it guarantees that $x$ always represents a 3D proposal in $\mathcal{M}$, and therefore takes the value of 1, which is omitted in Eq. (14). We have revised the description of this section to enhance its clarity and make it more understandable.
>
> To clarify this detail, we have updated the manuscript on lines 312-313.
>
> [F] Benedetto J J. Harmonic analysis and applications[M]. CRC Press, 2020.
>
> [G] Feller W. An introduction to probability theory and its applications, Volume 2[M]. John Wiley & Sons, 1991.
>
> > **Q5: The symbols in Eq. (11) are inconsistent with those in Figure 2. Eq (11) use ·  to denote matrix multiplication, but Figure 2 uses $\otimes$.**
> >
> A5: Thank you for pointing out this issue. We have corrected the notation in Figure 2 to align with the convention used in the equation. This change has been incorporated into the updated version of the paper to ensure clarity and consistency throughout.
>
> > **Q6: Minor issue**: Incorrect description in Figure 4(a): It appears that the target in Figure 4(a) is a table, not a couch, yet the descriptions for (a) and (b) are identical.
> >
>
> A6: Thank you for your valuable feedback. We have addressed this issue and made the necessary correction in Figure 4 in the updated version of the paper.
> > **Q7: In the category expert, if the class label can be directly extracted from the input text, why is a classifier needed for the text as well?**
> >
>
> A7: Thank you for your insightful question. A classifier is necessary in the category expert because the target words extracted by the natural language parser do not always correspond directly to predefined class labels. This occurs for several reasons:
>
> 1. **Synonyms and Variations**: Target words may act as synonyms for a category (e.g., "stool" and "chair") but are not explicitly mapped to the dataset's class labels.
> 2. **Ambiguity**: Some target words, such as "object," are too generic and cannot directly identify a specific category.
> 3. **Dataset Constraints**: The classification is limited by the dataset configuration, such as ScanNetV2, which defines only 20 broad categories, grouping many objects into an "others" class.
>
> These factors make direct use of parser-derived target words inaccurate, necessitating a trained classifier to accurately map textual inputs to the defined categories and ensure reliable performance in the 3D proposal classification process.

---

> ### Comment · Reviewer_tjsR · 2024-11-25
>
> Thanks for the authors' response and efforts. After reviewing the updated submission and response, I have some new concerns:
> 1. Implementation Details: The implementation details for methods [A-C] remain unclear. Providing detailed implementation information for these methods would help reviewers better evaluate the fairness of the performance comparisons.
> 2. Inconsistent Performance in Table 3: There are discrepancies in the performance reported in Table 3 between the old and new versions of the paper. Specifically, in the new version, the results for the “Unique” setting with 0.5 are identical to those for the “Multiple” setting with 0.25 in the old version.
> 3. In Multi-Expert Aggregation (MEA), the similarity score  $q$  between each proposal and the text is computed by aggregating the scores from three experts:  $p_{cls},  p_{attri},  p_{full}$. The authors state that "the aggregation method assigns greater weight to the inputs from more accurate experts" (Lines 318–319). However, according to Equation (3.14), the scores from different experts are multiplied and then normalized by dividing by the sum of the scores from all proposals. This process does not appear to involve explicit weighting to determine which expert is more accurate. Could the authors clarify how the method identifies and assigns greater weight to more accurate experts?
>
> I recommend addressing these issues to clarify and strengthen the paper.

---

> ### Author Response · Authors · 2024-11-27
> **Response to Reviewer tjsR (part-1)**
>
> > 1. Implementation Details: The implementation details for methods [A-C] remain unclear. Providing detailed implementation information for these methods would help reviewers better evaluate the fairness of the performance comparisons.
> >
>
> R1: We provide the implementation details and corresponding anonymous links for methods [A-C] as follows:
>
> **[A] OpenScene.**
>
> We selected the ***scannet_openseg_ensemble*** model version and weights based on the original paper and its official code repository. Following the approach described in 3D-STMN [1], we processed the data to generate instance masks corresponding to each description in the 3D-RES task, which were then used for evaluation.
>
> In the OpenScene inference setup, predicted masks are typically generated by calculating pointwise similarity between each point and the label set, followed by category classification. The most intuitive way to adapt this for 3D-RES is to compute the semantic similarity between the point cloud and the referring expression, filtering points based on a threshold to generate the predicted mask. However, the range of similarity scores between textual descriptions and target instances varies across examples, making it challenging to determine a universal threshold. To address this, we adopted a contrastive-inspired method: we replaced the category label of the target object in the original label set with the referring expression. This enabled pointwise category prediction and allowed us to extract the mask corresponding to the referring expression as the final prediction. The detailed implementation code is available at [run/evaluate_res.py](https://anonymous.4open.science/r/OpenScene_RES-DD2A/run/evaluate_res.py), along with the experimental results and metrics computation process, which can be found at [OpenScene_RES-DD2A](https://anonymous.4open.science/r/OpenScene_RES-DD2A). Please refer to [README_RES.md](https://anonymous.4open.science/r/OpenScene_RES-DD2A/README_RES.md) to run the code.
>
> **[B] OpenMask3D.**
>
> Firstly, we processed the data using the approach described in 3D-STMN [1], generating instance masks corresponding to each description in the 3D-RES task. These instance masks were then used for evaluation.
>
> Next, we followed the OpenMask3D setup. Specifically, we first computed class-agnostic masks and saved them. We then calculated mask features for each mask and stored these features. Using CLIP, we encoded the input textual descriptions to generate corresponding text features. The similarity between the text features and mask features was computed, and the mask with the highest similarity to the textual description was selected as the predicted output.
>
> The detailed implementation and testing code are available at [test.py](https://anonymous.4open.science/r/OpenMask3D_RES-D04F/test.py). Additionally, the experimental results are provided for reference at [OpenMask3D_RES-D04F](https://anonymous.4open.science/r/OpenMask3D_RES-D04F). Please refer to [README_RES.md](https://anonymous.4open.science/r/OpenMask3D_RES-D04F/README_RES.md) to run the code.
>
> **Checkpoints:**
>
> - **Mask Model:** scannet200_model
> - **Segment Anything Model:** ViT-H SAM model
> - **CLIP Model:** ViT-L/14@336px
>
> **[C] Open3DIS.**
>
> Firstly, we processed the data following the approach described in 3D-STMN [1], generating instance masks corresponding to each description in the 3D-RES task, which were subsequently used for evaluation purposes.
>
> Subsequently, we followed the setup of Open3DIS, processed the required data accordingly,  and executed the steps listed in RUN.md, specifically steps 1), 2), and 3), to obtain the 2D and 3D intermediate results. Since Open3DIS can, to some extent, be seen as an instance-aware improvement over OpenScene, we adapted the inference modifications for OpenScene, as outlined in [A], to better suit the 3D-RES task. Additionally, we further refined the process using the threshold filtering approach from Open3DIS to obtain the final predicted masks. The detailed implementation code can be found at [tools/referring_segmentation.py](https://anonymous.4open.science/r/Open3DIS_RES-E689/tools/referring_segmentation.py), and we have also provided the results from our experiments along with the metrics computation process for your reference, available at [Open3DIS_RES-E689](https://anonymous.4open.science/r/Open3DIS_RES-E689). Please refer to [README_RES.md](https://anonymous.4open.science/r/Open3DIS_RES-E689/README_RES.md) to run the code.
>
> We will update these details in the supplementary materials. Thank you for your suggestion, as it will enhance the reproducibility of our paper.
>
> [1] 3D-STMN: Dependency-Driven Superpoint-Text Matching Network for End-to-End 3D Referring Expression Segmentation. In AAAI 2024.

---

> ### Author Response · Authors · 2024-11-27
> **Response to Reviewer tjsR (part-2)**
>
> > 2. Inconsistent Performance in Table 3: There are discrepancies in the performance reported in Table 3 between the old and new versions of the paper. Specifically, in the new version, the results for the “Unique” setting with 0.5 are identical to those for the “Multiple” setting with 0.25 in the old version.
> >
>
> R2: We apologize for the confusion. The results in Table 3 of the updated manuscript were mistakenly filled in due to haste. We have revised the “Unique” results in Table 3 in the updated version. We are deeply grateful for your meticulous attention to detail, which has helped us ensure the accuracy and clarity of our work.
>
> > 3. In Multi-Expert Aggregation (MEA), the similarity score  between each proposal and the text is computed by aggregating the scores from three experts: $p_{cls}$, $p_{attri}$, $p_{full}$. The authors state that "the aggregation method assigns greater weight to the inputs from more accurate experts" (Lines 318–319). However, according to Equation (3.14), the scores from different experts are multiplied and then normalized by dividing by the sum of the scores from all proposals. This process does not appear to involve explicit weighting to determine which expert is more accurate. Could the authors clarify how the method identifies and assigns greater weight to more accurate experts?
> >
>
> R3: Thank you for your interest in our MEA module mechanism. In traditional MoE mechanisms, there is typically an expert selector that determines the weight of each expert. However, in a weakly supervised setting, we do not have sufficient labeled information to train such an expert selector as in traditional MoE systems. Therefore, we rethought an approach to expert selection without parameters, as you pointed out, by replacing addition with multiplication. The principle behind this is explained on page 3352, paragraphs 3–4, in [2]. To clarify why this mechanism automatically selects the experts with higher confidence, we will illustrate it with the following example.
>
> **Case:** For the referring expression “This is a brown wooden piano bench. It is in front of a piano. There is a brown tool box and an orange tool box next to the piano that the bench is in front of.” the probability distribution of three experts on the Ground Truth proposal and Distractor proposal is shown in Table a below:
>
> Table a. Example 1
>
> |  | $p_{cls}$ (High entropy, Low confidence) | $p_{attri}$ (High entropy, Low confidence) | $p_{full}$ (Low entropy, High confidence) |
> | --- | --- | --- | --- |
> | Distractor instance A (Brown Tool Box) | 0.45 | 0.4 | 0.05 |
> | Groundtruth instance B （Bench） | 0.3 | 0.25 | 0.3 |
>
> After applying the calculation in Equation 3.14, the probability weight for instance A (Brown Tool Box) is  $\frac{0.45\times 0.4 \times 0.05}{0.0090+0.0225}=0.286$, while the probability weight for instance B (Bench) is $\frac{0.3\times0.25\times0.3}{0.0090+0.0225}=0.714$, correctly selecting instance B as the ground truth.
>
> However, if we were to sum the values directly, as done in the “Average” strategy in the Table 4 MEA ablation experiment, $\frac{0.45+0.4+0.05}{3} > \frac{0.3+0.25+0.3}{3}$, which would result in incorrectly selecting instance A (Brown Tool Box).
>
> This mechanism consistently works because in high-confidence experts, the probability differences are more pronounced, as seen in the above table (0.05 vs. 0.3). The multiplication approach naturally gives higher weight to the expert with higher confidence, ensuring that its choice is prioritized. Therefore, this method allows us to select higher-confidence experts without the need for an expert selector, providing an effective solution for parameter-free MoE.
>
> We appreciate your question as it has helped clarify our contribution. We will include this discussion in the supplementary materials.
>
> [2] Hill T. Conflations of probability distributions[J]. Transactions of the American Mathematical Society, 2011, 363(6): 3351-3372.

---

> > ### Comment · Reviewer_tjsR · 2024-11-28
> >
> > Thanks for the authors' detailed response. I understand that there were some typos in the earlier manuscript, and all of my concerns have now been addressed. Based on this, I will raise my rating to borderline accept.

---

> ### Author Response · Authors · 2024-11-28
> **Response to Reviewer tjsR**
>
> Thank you for your positive response. Your constructive feedback has greatly improved our work.

---

### Official Review · Reviewer_wNd8 · 2024-11-01

**Soundness:** 4
**Presentation:** 3
**Contribution:** 3
**Rating:** 8
**Confidence:** 4

**Summary:**

The authors present a weakly supervised 3D-RES task and the Multi-Expert Network (MEN), an innovative framework that aims to mitigate the reliance on costly annotations by leveraging diverse complementary visual-language cues. MEN consists of two core modules: the Multi-Expert Mining (MEM) module and the Multi-Expert Aggregation (MEA) module. The MEM module employs multiple visual-language experts to independently mine distinct visual and linguistic cues, aligning visual and textual contexts from three complementary perspectives. Meanwhile, the MEA module synthesizes the outputs of these experts through a distribution-based fusion approach, adaptively assigning appropriate fusion weights to enhance integration. Extensive experiments on the ScanRefer and Multi3DRefer datasets underscore the efficacy of this approach for the weakly supervised 3D-RES task.

**Strengths:**

1、	The high cost of obtaining annotations in the 3D domain has been a significant barrier to progress in related fields. Thus, this work’s proposal of a new weakly supervised 3D-RES task, aiming to reduce dependency on 3D annotations, is highly valuable and offers a fresh direction for advancing the community's development.

2、	Additionally, this paper introduces a novel weakly supervised 3D-RES framework, MEN, to address the weakly supervised 3D-RES task, providing a strong baseline. The incorporation of a multi-expert fusion approach into the 3D-RES task is also innovative and noteworthy.

3、	The proposed MEM module in this paper effectively explores different visual-language relationships by leveraging three complementary cues: Full-context, Attribute, and Category.

**Weaknesses:**

1、	Lines 202-205 lack a detailed and clear description of visibility. Additionally, some symbol notations in the text are ambiguous; for instance, the "textual full-context feature" mentioned in line 215 is referred to as "2D-fine text features" in line 199.

2、	The proposed method demonstrates high performance on traditional 3D-RES tasks; however, it exhibits lower performance on certain metrics in the 3D-GRES task, particularly in zero-target scenarios. What factors contribute to this underperformance, and are there potential improvements that could be implemented?

3、	This two-stage paradigm is influenced by the proposals generated in the first stage. Therefore, how would the overall performance of the network change if a different instance segmentation model were utilized in the first stage?

4、	Furthermore, I would like to see the performance upper bound of the network when errors from the first-stage segmentation model are excluded.

**Questions:**

1、	This paper utilizes an off-the-shelf natural language parser to extract attribute-level text, which can be understood as a standard approach. However, I am quite curious whether employing a powerful large language model (LLM) to construct attribute phrases as text inputs for the attribute experts could further enhance the model's performance.

---

> ### Author Response · Authors · 2024-11-24
> **Official Comment by Authors (part-1)**
>
> > **Q1: Lines 202-205 lack a detailed and clear description of visibility. Additionally, some symbol notations in the text are ambiguous; for instance, the "textual full-context feature" mentioned in line 215 is referred to as "2D-fine text features" in line 199.**
> >
> A1: Thank you for pointing out these issues. We have first clarified **visibility**, defining it as the total number of projected points of a 3D proposal visible in a given frame [A]. A detailed description of visibility and its computation is now included in the lines 994-1021 of Section F in the updated version. Additionally, we will review and refine **symbol notations** to ensure clarity and consistency throughout the text, focusing on the discrepancies you mentioned.
>
> [A] Open-YOLO 3D: Towards Fast and Accurate Open-Vocabulary 3D Instance Segmentation. In Arxiv 2024.
>
> > **Q2: The proposed method demonstrates high performance on traditional 3D-RES tasks; however, it exhibits lower performance on certain metrics in the 3D-GRES task, particularly in zero-target scenarios. What factors contribute to this underperformance, and are there potential improvements that could be implemented?**
> >
>
> A2: Thanks for your insightful question. Our analysis found key factors causing underperformance in zero-target scenarios. We found that higher thresholds improve performance in zero-target scenarios but decrease performance in multi-target scenarios. Details are in Table 10 of the supplementary materials. Thus, using a simple threshold is not effective for the 3D-GRES task.
>
> To address this, we develop new methods to handle zero-target cases better. We treat zero-target detection as a classification task. This aligns with our weakly supervised setup, which uses only category labels without adding expensive mask labels. We introduce a separate *zt_head* to predict if an instance is a zero target. The *zt_head* is trained using zero-target labels as supervision, as explained in section D of the updated paper. The results, shown in the table below, show that adding the *zt_head* significantly improved performance in zero-target cases, without affecting performance in other scenarios.
>
> We appreciate your valuable suggestion and have included this adjustment in the next version (highlighted in blue, lines 365–367, 429-431, 945-977). This enhancement enriches the content and improves the model's robustness. Thank you again for your thoughtful input.
>
> Table a: Results of 3D-GRES, including mIoU and Acc\@0.5, where "w/o zt_head" means that the zt_head was not used, and "w/ zt_head" means that the zt_head was used to predict whether it is a zero target.
>
> |  | mIoU | zt  w/  dis | zt  w/o  dis | st  w/ dis | st  w/o dis | mt  | Overall |
> | --- | --- | --- | --- | --- | --- | --- | --- |
> | w/o zt_head | 30.6 | 1.3 | 0.4 | 28.9 | 59.5 | 19.8 | 30.1 |
> | w/ zt_head | 33.4 | 22.2 | 55.9 | 28.7 | 59.4 | 19.7 | 33.4 |
>
> > **Q3: This two-stage paradigm is influenced by the proposals generated in the first stage. Therefore, how would the overall performance of the network change if a different instance segmentation model were utilized in the first stage?**
> >
>
> A3: Thank you for your valuable suggestion.  In response, we have updated our experiments as per your recommendation and presented the results in Table b, where we replaced the instance segmentation model with SPFormer[B]. We have included this part in Table 13 in the updated version. The results indicate that the substitution of different instance segmentation models has little effect on the performance of our model, which further underscores its robustness.
>
> [B] Superpoint Transformer for 3D Scene Instance Segmentation. In AAAI 2023.
>
> Table b: Results of 3D-RES with different instance segmentation models: SoftGroup and SPFormer.
>
> |  | mIoU | Unique Acc\@0.25 | Unique Acc\@0.5 | Multiple Acc\@0.25 | Multiple Acc\@0.5 | Overall Acc\@0.25 | Overall Acc\@0.5 |
> | --- | --- | --- | --- | --- | --- | --- | --- |
> | SoftGroup | 33.4 | 83.1 | 79.1 | 32.6 | 27.9 | 42.4 | 37.8 |
> | SPFormer | 34.2 | 81.6 | 77.9 | 31.2 | 26.8 | 41.0 | 36.7 |
> > **Q4: Furthermore, I would like to see the performance upper bound of the network when errors from the first-stage segmentation model are excluded.**
> >
>
> A4: Thank you for your constructive suggestion. To assess the network's performance without errors from the first-stage segmentation, we used instance-level GT masks and GT class labels as inputs, resulting in an mIoU of 48.1. This part has been included in Table 13 in the updated version.
>
> Table c: Exploration of upper bound when using the GroundTruth Instance labels.
>
> |  | mIoU | Unique Acc\@0.25 | Unique Acc\@0.5 | Multiple Acc\@0.25 | Multiple Acc\@0.5 | Overall Acc\@0.25 | Overall Acc\@0.5 |
> | --- | --- | --- | --- | --- | --- | --- | --- |
> | SoftGroup | 33.4 | 83.1 | 79.1 | 32.6 | 27.9 | 42.4 | 37.8 |
> | GT Ins. | 48.1 | 89.2 | 89.2 | 38.1 | 38.1 | 48.1 | 48.1 |

---

> ### Author Response · Authors · 2024-11-24
> **Official Comment by Authors (part-2)**
>
> > **Q5: This paper utilizes an off-the-shelf natural language parser to extract attribute-level text, which can be understood as a standard approach. However, I am quite curious whether employing a powerful large language model (LLM) to construct attribute phrases as text inputs for the attribute experts could further enhance the model's performance.**
> >
>
> A5: Thank you for your insightful suggestion. In response, we have expanded the discussion on LLMs, and the results are now presented in Table d. We utilized GPT-3.5-turbo to generate attribute phrases, with the input prompts displayed below. The results suggest that while LLMs are able to generate more accurate attribute phrases, their effect on the overall model performance remains minimal. We have included this part in Line 880-917 of Section B in the updated version.
>
> Prompts:
>
> ```json
> {
>     "role":"system",
>     "content":
>         "You are an intelligent chatbot designed to extract attribute information of a target object from a given sentence."
>         "Your task is to analyze the sentence I provide, which describes a target object, and distill it into a concise descriptive phrase containing the target object's name and its attributes."
>         "Note that all attribute words must be directly taken from the input description without alteration, omission, or addition."
>         "Furthermore, you must not use attribute words unrelated to the target object in the description."
> },
> {
>     "role":"user",
>     "content":
>         "Please extract the target and its attributes from the following input text and output a concise phrase."
>         f"Input sentence: {text}\n"
>         "DO NOT PROVIDE ANY OTHER OUTPUT TEXT OR EXPLANATION. Only provide the target attribute phrase."
>         "------"
>         "Here are some examples:"
>         "Input sentence: There is a brown wooden chair. Placed beside other chairs in the middle of the kitchen."
>         "Your response should look like this: A brown wooden chair."
>         "Input sentence: The couch is north of the round coffee table. the couch has three seats and two armrests."
>         "Your response should look like this: A couch with three seats and two armrests."
>         "Input sentence: There is a dark brown chair. brown leather and placed in he kitchen table."
>         "Your response should look like this: A dark brown wooden and leather chair."
> }
> ```
>
> Table d: Performance Comparison of Attribute Phrases Generated by the Parser and LLM.
>
> |  | mIoU | Unique Acc\@0.25 | Unique Acc\@0.5 | Multiple Acc\@0.25 | Multiple Acc\@0.5 | Overall Acc\@0.25 | Overall Acc\@0.5 |
> | --- | --- | --- | --- | --- | --- | --- | --- |
> | Parser | 33.4 | 83.1 | 79.1 | 32.6 | 27.9 | 42.4 | 37.8 |
> | LLM | 33.5 | 83.4 | 79.4 | 32.7 | 27.9 | 42.6 | 37.9 |

---

> > ### Comment · Reviewer_wNd8 · 2024-11-28
> > **Official Comment by Reviewer wNd8**
> >
> > I appreciate the authors' detailed and thorough responses, as well as their efforts in incorporating my suggestions into the paper. The authors have effectively addressed the zero-target problem through a straightforward zero-target head design and conducted comprehensive visual-side ablations across multiple backbones, further exploring performance upper bounds. All my concerns have been fully addressed.
> > Furthermore, I reviewed the discussions between the authors and other reviewers. I am convinced that, as the first work to propose this task, the level of completion achieved by this study is already impressive, offering significant insights to the research community. Such explorations and contributions should be encouraged to stimulate innovation and progress within the field.
> > As a result, I maintain my original score of 8.

---

> ### Author Response · Authors · 2024-11-28
> **Response to Reviewer wNd8**
>
> Thank you for your positive response. We are very glad that our reply meets your expectations. We will continue to update and refine our paper accordingly.

---

### Author Response · Authors · 2024-11-24
**Common response**

We would like to express our gratitude to the reviewers for their valuable feedback and positive comments on our paper. Their thoughtful and detailed evaluations have been instrumental in enhancing both the clarity and overall quality of our work.

We appreciate Reviewer **wNd8** $\color{red}{(Rating:\mathbf{8},\ Confidence:\mathbf{4})}$ for acknowledging the strengths of our paper. Specifically, they recognized the contribution of our proposed weakly supervised 3D-RES task in advancing the community's development. They also appreciated the incorporation of a multi-expert fusion approach into the 3D-RES task. Additionally, they acknowledge that our MEM  module effectively explores different visual-language relationships by leveraging three complementary cues.

Reviewer **tjsR** $\color{red}{(Rating:\mathbf{5},\ Confidence:\mathbf{4})}$ acknowledged our extensive comparative experiments with state-of-the-art methods and highlighted our method's effectiveness in accurately segmenting 3D objects based on text descriptions.

Reviewer **kMXL** $\color{red}{(Rating:\mathbf{6},\ Confidence:\mathbf{5})}$  recognized the novelty of our weakly supervised 3D-RES framework and the effectiveness of the Multi-Expert Network in leveraging multiple cues for referring expression reasoning. They also commended the clarity and accessibility of our writing. Additionally, we are grateful for their acknowledgment of the significance of our experimental results, emphasizing their role in advancing weakly supervised 3D-RES research.

Reviewer **nJzX** $\color{red}{(Rating:\mathbf{5},\ Confidence:\mathbf{4})}$ expressed appreciation for the clear articulation of our motivation and the thorough ablation studies conducted on our modules. They also acknowledged the potential of our proposed weakly supervised 3D-RES framework to eliminate the need for labor-intensive point-level mask annotations.

We sincerely appreciate the reviewers for recognizing these strengths. We genuinely appreciate the time and effort they have dedicated to this process. Their comments have further motivated us to address the concerns and improve the weaknesses pointed out in their reviews. We are committed to addressing their concerns and providing a detailed response in our rebuttal.

---

### Meta-Review · Area_Chair_NtrQ · 2024-12-22

**Metareview:**

This paper received significantly mixed reviews. The reviewers recognized that the proposed research direction, i.e., weakly supervised 3D referring expression segmentation (RES), is new and practically valuable, the proposed framework is novel and shows strong performance in the task. They at the same time raised concerns with underperformance on zero-target or multi-target scenes (wNd8, kMXL, nJzX), sensitivity to the instance segmentation model for proposal generation (wNd8, nJzX), lack of discussions of and comparisons with open-vocabulary 3D segmentation models that can handle the proposed task to some extent (kMXL, nJzX), inability to handle cross-frame objects (kMXL, nJzX), insufficient ablation study (kMXL), and incremental technical innovation (nJzX). Also, all the reviewers pointed out clarity issues.

The authors' rebuttal and subsequent responses in the discussion period address some of these concerns but failed to fully assuage all of them, even after a large body of author-reviewer discussions. As a result, after the discussion period, the two negative reviewers (kMXL, nJzX) still have concerns about weak clarity of the revised manuscript, the issue in handling cross-frame objects, sensitivity to the instance segmentation model, the low performance on zero-target or multi-target scenes, and lack of technical innovation. The AC found that these comments are all valid, although some of them, such as the underperformance issue, can be execused as this work is the first attempt to weakly supervised 3D RES.

In particular, the clarity issue should be addressed appropriately before publication. Although the AC sincerely appreciates the extensive additional experiments conducted during the rebuttal period and the latest revision including their results, their presentation in the current manuscript is not well organized. To ensure that the paper is a good first introduction to weakly supervised 3D RES, it is essential to provide a comprehensive and well-degisned benchmark for the task. In this regard, the paper should elaborate on all datasets, evaluation settings, and their purposes (e.g., evaluating conventional 3D RES, generalized 3D RES, insensitivity to proposal generator) *in the main text*, not in the appendix, and the results of all the experiments should be analyzed collectively. The AC sees that the volume of revision required for this purpose will exceed what is typically expected from a camera-ready revision. In addition, other remaining concerns with cross-frame objects and sensitivity to proposal generator should be carefully addressed to further strenghthen the paper of course.

Putting these together, the AC considers that the remaining concerns outweigh the positive comments and rebuttal, and thus regrets to recommend rejection. The authors are strongly encouraged to carefully revise the paper to reflect the valuable comments by the reviewers, to add new results brought up in the rebuttal and discussions, and to further improve the quality of writing for next submissions.

**Additional Comments On Reviewer Discussion:**

- **Missing analytics for low performance in zero- and multi-target scenes (wNd8, kMXL, nJzX)**: wNd8's concern has been fully assuaged, but the two negative reviewers, kMXL and nJzX, still pointed out this issue after the discussion period. The AC found that the rebuttal is more convincing than the comments by the negative reviewers. RES in zero- or multi-target scenes, so called generalized RES, has been known to be substantially challenging in the field of RES for images. Also, the reviewers consider introducing a zero-target classification head as a negative point, but it has been investigated as a simple yet effective solution to the task recently in the literature. Hence, *the AC did not weigh this issue heavily when making the final decision*.
- **Sensitivity to the instance segmentation model for proposal generation  (wNd8, nJzX)**: The authors addressed this concern by presenting additional experiments and analyses, which demonstrate that the proposed framework is not very sensitive to the instance segmentation model used for proposal generation, and that its performance is not very far from the upperbound using groundtruth proposals. However, the concern of Reviewer nJzX has not been well assuaged since he or she believes that this result contradicts to the main goal of the paper, i.e., reducing annotation across different environments. The AC sees that the reviewer's comment is valid; *this is definitely a negative point of the paper*. To be specific, it means potential data leakage as commented by the reviewer, and over-reliance on the proposal generator that is not a contribution of this paper.
- **Inability to handle cross-frame objects (kMXL, nJzX)**: This is clearly a limitation of the paper; the authors and even the most supportive reviewer admitted. However, *the AC believes this is not a ground for rejection* regarding, again, that this paper is the first attempt to weakly supervised 3D RES and that the issue has been originated from limitations of existing datasets. Also, the authors may define a subset of the validation split that includes only such challenging examples to evaluate the capability to handle cross-frame objects of follow-up studies. Of course, the authors are strongly encouraged to address this issue carefully to strengthen the paper, especially in terms of benchmark, before publication.
- **Clarity issues (wNd8, tjsR, kMXL, nJzX)**: *These are one of the main objections of the AC*. Although the AC sincerely appreciates the extensive additional experiments conducted during the rebuttal period and the latest revision including their results, their presentation in the current manuscript is not well organized, as commented by Reviewer nJzX. To ensure that the paper is a good first introduction to weakly supervised 3D RES, it is essential to provide a comprehensive and well-degisned benchmark for the task. In this regard, the paper should elaborate on all datasets, evaluation settings, and their purposes (e.g., evaluating conventional 3D RES, generalized 3D RES, insensitivity to proposal generator) in the main text, not in the appendix, and the results of all the experiments should be analyzed collectively. The AC sees that *the volume of revision required for this purpose will exceed what is typically expected from a camera-ready revision*.
- **Lack of discussions of and comparisons with open-vocabulary 3D segmentation methods that are capable of handling the proposed task to some extent (kMXL, nJzX)**: The authors successfully assuaged this concern by the revision and additional experimental results, the latter clearly demonstrate that the proposed method outperforms open-vocabulary 3D segmentation in weakly supervised 3D referring expression segmentation. The reviewers first were not fully assuaged due to the significantly low performance of the open-vocabulary models but the authors' subsequent responses sound fairly convincing--that is due to the difference between the tasks.
- **Incremental technical innovation (nJzX)**: The authors' response to this comment sounds reasonable and the reviewer did not mention it in the end. However, the AC still feels that the technical innovation is limited as the proposed method is a combination of existing ideas and techniques (including the theoretical explanation of the proposed aggregation strategy), some of which are not directly from the 3D RES literature though. *This may not be a reason for rejection, but it does make the AC question whether this paper is a good fit for ICLR.

---

### Decision · Program_Chairs · 2025-01-22

Reject